DOI: 10.1038/s41467-018-02897-7　　**OPEN**

# Gimap5-dependent inactivation of GSK3β is required for CD4+ T cell homeostasis and prevention of immune pathology

Andrew R. Patterson [1,2], Mehari Endale [3], Kristin Lampe[1], Halil I. Aksoylar[4], Aron Flagg[5], Jim R. Woodgett [6], David Hildeman [1,2], Michael B. Jordan[1,2], Harinder Singh[1,2], Zeynep Kucuk[7], Jack Bleesing[7] & Kasper Hoebe[1,2,8]

GTPase of immunity-associated protein 5 (Gimap5) is linked with lymphocyte survival, autoimmunity, and colitis, but its mechanisms of action are unclear. Here, we show that Gimap5 is essential for the inactivation of glycogen synthase kinase-3β (GSK3β) following T cell activation. In the absence of Gimap5, constitutive GSK3β activity constrains c-Myc induction and NFATc1 nuclear import, thereby limiting productive CD4+ T cell proliferation. Additionally, Gimap5 facilitates Ser389 phosphorylation and nuclear translocation of GSK3β, thereby limiting DNA damage in CD4+ T cells. Importantly, pharmacological inhibition and genetic targeting of GSK3β can override Gimap5 deficiency in CD4+ T cells and ameliorates immunopathology in mice. Finally, we show that a human patient with a *GIMAP5* loss-of-function mutation has lymphopenia and impaired T cell proliferation in vitro that can be rescued with GSK3 inhibitors. Given that the expression of Gimap5 is lymphocyte-restricted, we propose that its control of GSK3β is an important checkpoint in lymphocyte proliferation.

[1] Division of Immunobiology, Cincinnati Children's Hospital Research Foundation, 3333 Burnet Avenue, Cincinnati, OH 45229, USA. [2] Immunology Graduate Program, Cincinnati Children's Hospital Medical Center and the University of Cincinnati College of Medicine, 231 Albert Sabin Way # E251n, Cincinnati, OH 45267, USA. [3] Division of Neonatology and Pulmonary Biology, Cincinnati Children's Hospital Research Foundation, 3333 Burnet Avenue, Cincinnati, OH 45229, USA. [4] Department of Genetics and Complex Diseases, Harvard T.H. Chan School of Public Health, 677 Huntington Avenue, Boston, MA 02115, USA. [5] Pediatric Hematology/Oncology and Blood & Marrow Transplant, Cleveland Clinic Children's, 9500 Euclid Avenue, Cleveland, OH 44195, USA. [6] The Lunenfeld-Tanenbaum Research Institute, Mount Sinai Hospital, 600 University Avenue, Toronto, ON M5G 1X5, Canada. [7] Division of Bone Marrow Transplantation & Immune Deficiency, Cincinnati Children's Hospital Research Foundation, 3333 Burnet Avenue, Cincinnati, OH 45229, USA. [8] Department of Pediatrics, University of Cincinnati, College of Medicine, 3230 Eden Avenue, Cincinnati, OH 45267, USA. Correspondence and requests for materials should be addressed to K.H. (email: kasper.hoebe@cchmc.org)

GTPase of immunity-associated protein 5 (Gimap5) is linked with lymphocyte survival, immune homeostasis, and (auto)immune disease. Specifically, polymorphisms in human GIMAP5 are associated with increased risk of islet autoimmunity in type 1 diabetes (T1D), systemic lupus erythematosus (SLE)[1–3], and asthma[4]. Mice and rats with complete loss-of-function (LOF) mutations have reduced lymphocyte survival, loss of immunological tolerance predisposing to autoimmunity and colitis, and abnormal liver pathology resulting from persistent post-natal extramedullary hematopoiesis[5–14]. Despite this critical role of Gimap5 in lymphocyte survival and peripheral tolerance, the underlying mechanism(s) are unclear.

Gimap proteins are predominantly expressed in lymphocytes and regulate lymphocyte survival during development, selection, and homeostasis[15]. Members of this family share a GTP-binding AIG1 homology domain[16,17] and seem to be localized to different subcellular compartments, with Gimap5 localizing in multi-vesicular bodies (MVB) and lysosomes[18]. Overall, a function for Gimaps in maintaining T cell homeostasis is not clearly defined. We previously generated Gimap5-deficient mice, so-called "sphinx" mice, which have a missense mutation in Gimap5 that results in what is essentially a null allele[6]. $Gimap5^{sph/sph}$ mice progressively lose CD4$^+$ T cells and B cells, an effect that is associated with reduced regulatory T (Treg) cell function, while remaining CD4$^+$ T cells have an activated phenotype, but have an impaired capacity to proliferate[5,6]. These immunologic defects result in spontaneous and lethal colitis that is preventable with CD4$^+$ T cell depletion, Treg cell transplantation, or antibiotic therapy[5,6]. Despite these effective therapies, the cell-intrinsic defects in $Gimap5^{sph/sph}$ CD4$^+$ T cells, including their reduced survival, persist. In addition to colitis, livers from $Gimap5^{sph/sph}$ mice have an abnormal morphology with extramedullary hematopoiesis and associated foci of hematopoietic cells and hepatocyte apoptosis[6–8].

The family of glycogen synthase kinases-3 (GSK3) includes constitutively active protein serine/threonine kinases encoded by two genes, Gsk3a and Gsk3b. Studies have shown that GSK3β has an essential function in T cell differentiation and proliferation[19–22]. GSK3 phosphorylates a variety of substrates, often within the phosphodegron domains, thereby regulating protein ubiquitination/degradation and their activity[23]. Among the substrates targeted directly by GSK3 are c-Myc, NFATc1, Mcl-1, and β-catenin[24]. Studies have established that upon antigen-specific activation of T cells, GSK3β activity is inhibited[19–22], thereby facilitating T cell activation and proliferation. Phosphorylation of GSK3β at residues Ser9 or Ser389 and physical separation of GSK3β from its target proteins through vesicle association have been proposed as mechanisms of GSK3β inhibition[25,26]. Inhibition of GSK3 by phosphorylation of Ser389 is essential for lymphocyte viability upon double-stranded DNA breaks (DSB) evident during V(D)J recombination in thymic T cell development or in B cells undergoing immunoglobulin class switch recombination (CSR)[27]. Further reports have demonstrated that GSK3β is involved in the regulation of nonhomologous end-joining repair of DSB, with pharmacological inhibition and shRNA knockdown of GSK3β promoting DNA repair[28]. Nonetheless, the exact mechanism by which inhibition of GSK3 activity is regulated in T cells is unclear.

In the current study, we provide an important insight into the functional role of Gimap5. Gimap5 is a critical inhibitor of GSK3β in both human and mouse CD4$^+$ T cells, affecting c-Myc and Mcl-1 expression, NFATc1 nuclear translocation, and T cell fitness by controlling the DNA damage response occurring during T cell proliferation.

## Results

**$Gimap5^{sph/sph}$ mice have normal thymic output of CD4$^+$ T cells.** Studies implicate a loss of peripheral CD4$^+$ T cells in both Gimap5-deficient mice and rats[6,8,12,15,29–31]. To determine whether the observed reduction in peripheral CD4$^+$ T cells might stem from abnormal thymic CD4$^+$ T cell development, we investigated whether the survival and/or output of thymic CD4$^+$ T cells in $Gimap5^{sph/sph}$ mice was affected. To assess the survival of thymocytes, we isolated thymic CD4$^+$ T cells and cultured them in the presence of IL-7 for 1 week. Subsequently, the number of live single positive (SP) CD4$^+$/CD8$^-$ T cells was quantified at various incubation times. Notably, our data indicate no differences in the survival ex vivo between SP CD4$^+$ thymocytes isolated from wildtype (WT) and $Gimap5^{sph/sph}$ mice (Supplementary Figure 1A).

We next assessed if reduced thymic output of CD4$^+$ T cells might contribute to lymphopenia in $Gimap5^{sph/sph}$ mice, and quantified the presence of recent thymic emigrants (RTE)[32] in the spleen of WT and $Gimap5^{sph/sph}$ mice. Importantly, we found no marked differences in the frequency of splenic RTE as defined by CD24$^{hi}$ CD4$^+$ T cells between 3-week-old WT or $Gimap5^{sph/sph}$ mice (Supplementary Figure 1B). These data are in line with our previous studies showing $Gimap5^{sph/sph}$ mice have a relatively normal thymic development of CD4$^+$ T cells[6].

**Activation-induced cell death of peripheral CD4$^+$ T cells.** We next focused on the peripheral survival of CD4$^+$ T cells in $Gimap5^{sph/sph}$ mice. We considered that either post-thymic survival of $Gimap5^{sph/sph}$ CD4$^+$ T cells or T-cell receptor (TCR)-induced activation contributes to the loss of CD4$^+$ T cells in the periphery. The latter would be consistent with our previous studies showing that $Gimap5^{sph/sph}$ T cells failed to proliferate after TCR stimulation with αCD3/αCD28 ± IL-2[6]. Moreover, a progressive loss of CD4$^+$ T cells is observed post-weaning—a period in which the CD4$^+$ T cell compartment has to cope with marked changes in gut microbial antigens. To directly test the role of TCR activation in vivo, we generated $Gimap5^{sph/sph}$; $Rag2^{-/-}$;OT-II TCR transgenic mice. These mice contain CD4$^+$ T cells that only recognize the OVA$_{323–339}$ epitope presented by I-Ab in C57BL/6 mice and lack endogenous CD4$^+$ T cell repertoires and CD8$^+$ T cells and B cells. Interestingly, their CD4$^+$ T cell compartment is largely maintained in the absence of antigen and predominantly have a naïve (CD44$^{lo}$;CD62$^{hi}$) phenotype (Fig. 1a, b). In contrast, $Gimap5^{sph/sph}$;$Rag2^{-/-}$;OT-II mice exposed to ovalbumin in drinking water have a reduced CD4$^+$ T cell population with an increased proportion of remaining CD4$^+$ T cells displaying a memory-like phenotype (CD44$^{hi}$;CD62L$^{lo}$) compared to WT controls (Fig. 1a, b). Moreover, these mice failed to induce antigen-specific Treg cells (Fig. 1c). These data show that antigen-specific activation of $Gimap5^{sph/sph}$ CD4$^+$ T cells directly contributes to the loss of these cells in vivo. To assess the potential contribution of reduced homeostatic survival of peripheral CD4$^+$ T cells, we isolated CD4$^+$ T cells from the spleen of WT and $Gimap5^{sph/sph}$ mice and cultured them in the presence of IL-7. The number of live CD4$^+$ T cells was quantified at various time points; in contrast to the thymic SP CD4 T cells, splenic $Gimap5^{sph/sph}$ CD4$^+$ T cell numbers were significantly reduced compared to WT (Fig. 1d). These data suggest that peripheral $Gimap5^{sph/sph}$ CD4$^+$ T cells have a reduced peripheral survival compared to WT CD4$^+$ T cells.

We next sought to identify whether the cell-intrinsic apoptosis pathways contributed to the reduced CD4$^+$ T cell survival in $Gimap5^{sph/sph}$ mice. One study suggests that Gimap5 directly binds anti-apoptotic proteins Bcl-2 and Bcl-xL[15], while another suggests that loss of Gimap5 is associated with increased Mcl-1

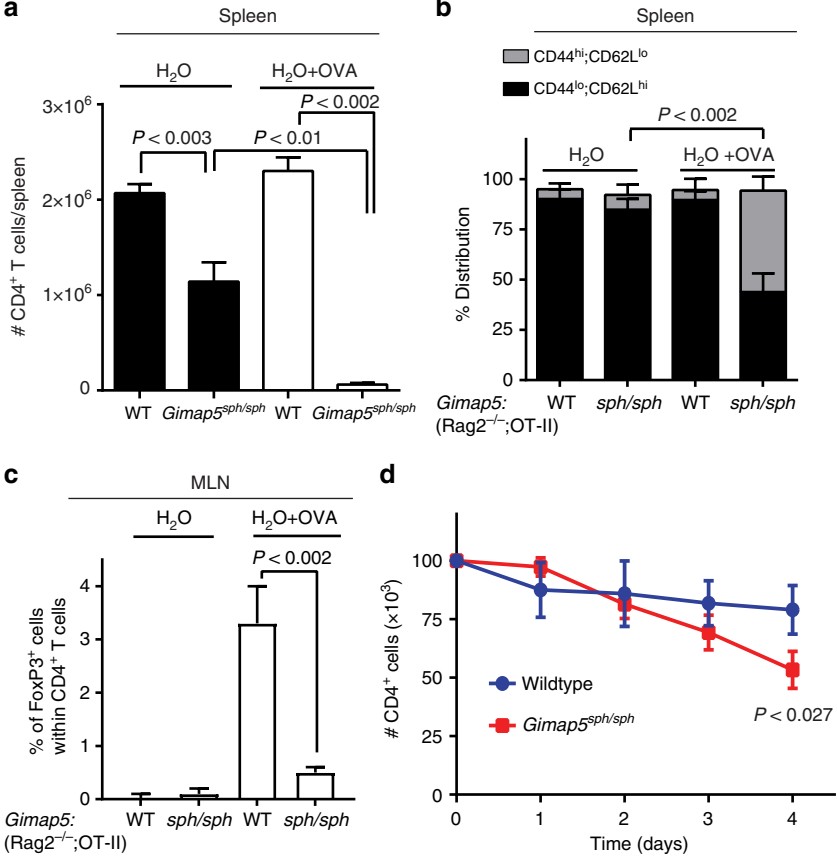

**Fig. 1** Loss of Gimap5 impairs CD4$^+$ T cell survival and iTreg cell induction. Ten-week-old control $Gimap5^{WT/WT}$;$Rag2^{-/-}$;OT-II and $Gimap5^{sph/sph}$;$Rag2^{-/-}$; OT-II mice received either normal water or water containing 1 mg/mL ovalbumin ad libitum during a 5-week period. At 15 weeks of age, **a** the number of splenic CD4$^+$ T cells and **b** the percentage of naïve (CD44$^{lo}$;CD62$^{hi}$) and memory-like (CD44$^{hi}$;CD62$^{lo}$) CD4$^+$ T cells was determined using flow cytometry. **c** Frequency of iTregs (CD4$^+$CD25$^+$) in the mesenteric lymph nodes (MLN) of vehicle- and OVA-treated mice ($n = 4$). **d** Survival of peripheral (splenic) CD4$^+$ T cells ex vivo in the presence of IL-7 (5 ng/mL) as determined by live/dead staining and flow cytometry ($n = 3$). Data represent mean values + SD of samples from individual mice; statistical significance is determined by Student's two-tailed test

degradation and compromised mitochondrial integrity in hematopoietic progenitor cells[7]. To test the potential role of Bcl-2 and Bcl-xL in the reduced survival of $Gimap5^{sph/sph}$ CD4$^+$ T cells, we crossed the $Gimap5^{sph/sph}$ allele onto a $Bim^{-/-}$ background. Importantly, we observed no rescue of peripheral T cell survival (Supplementary Figure 2A–D). We next assessed Mcl-1 expression in resting and activated CD4$^+$ T cells and observed that accumulation of Mcl-1, similar to hematopoietic progenitor cells[7], was reduced in $Gimap5^{sph/sph}$ CD4$^+$ T cells compared to WT controls after 24 h of stimulation with αCD3/αCD28 (Supplementary Figure 2E). Studies have shown that $Bax/Bak$ deletion can prevent CD4$^+$ T cell death in the complete absence of Mcl-1[33]. However, crossing the $Gimap5^{sph/sph}$ allele onto a $Bax/Bak$ double-deficient background failed to rescue CD4$^+$ T cell survival (Supplementary Figure 2F), thus the impact of the reduced Mcl-1 expression is unclear. Finally, we sought to investigate if Gimap5-deficient CD4$^+$ T cells were more susceptible to cell-extrinsic apoptosis pathways. Stimulation of CD4$^+$ T cells with αCD3/αCD28 in the absence/presence of activating Fas antibodies for 8 h, however, resulted in similar frequencies of apoptotic and dead cells between WT and $Gimap5^{sph/sph}$ CD4$^+$ T cells (Supplementary Figure 2G).

Together these data indicate that the reduced survival of peripheral CD4$^+$ T cells in $Gimap5^{sph/sph}$ mice occurs independently of the cell-intrinsic apoptosis pathways.

**$Gimap5^{sph/sph}$ CD4$^+$ T cells have abnormal GSK3 activity**. To identify the molecular defects that mediate decreased proliferation, we assessed the activation of proximal signaling pathways in CD4$^+$ T cells from 3-week-old $Gimap5^{sph/sph}$ mice. At this age, CD4$^+$ T cell numbers are comparable to WT with a normal frequency of naïve CD4$^+$ T cells that have a quiescent phenotype (Supplementary Figure 3A). Importantly, our data indicated no abnormalities in the activation of proximal signaling pathways (IκB, ERK, JNK, AKT, p38, mTORC1, and p70 S6K) (Supplementary Figure 3B–D). However, we observed a marked reduction in protein levels of the transcription factor (TF) c-Myc in activated $Gimap5^{sph/sph}$ CD4$^+$ T cells (Fig. 2a). C-Myc is a TF necessary for the metabolic programming of T cells following activation and is required for optimal T cell proliferation[34,35]. Notably, $myc$ mRNA levels in resting or αCD3/αCD28-stimulated $Gimap5^{sph/sph}$ CD4$^+$ T cells were comparable to WT controls, suggesting that reduction in protein levels was the result of changes in post-translational regulation (Fig. 2b). Post-translational regulation of c-Myc is mediated in part by the family of GSK3 proteins, which is comprised of constitutively active protein serine/threonine kinase paralogs GSK3α and GSK3β. GSK3α/β phosphorylate c-Myc at Thr58, priming it for ubiquitination and subsequent proteasomal degradation[36].

Studies suggest that GSK3β plays an essential role in T cell differentiation and proliferation. Specifically, upon antigen-

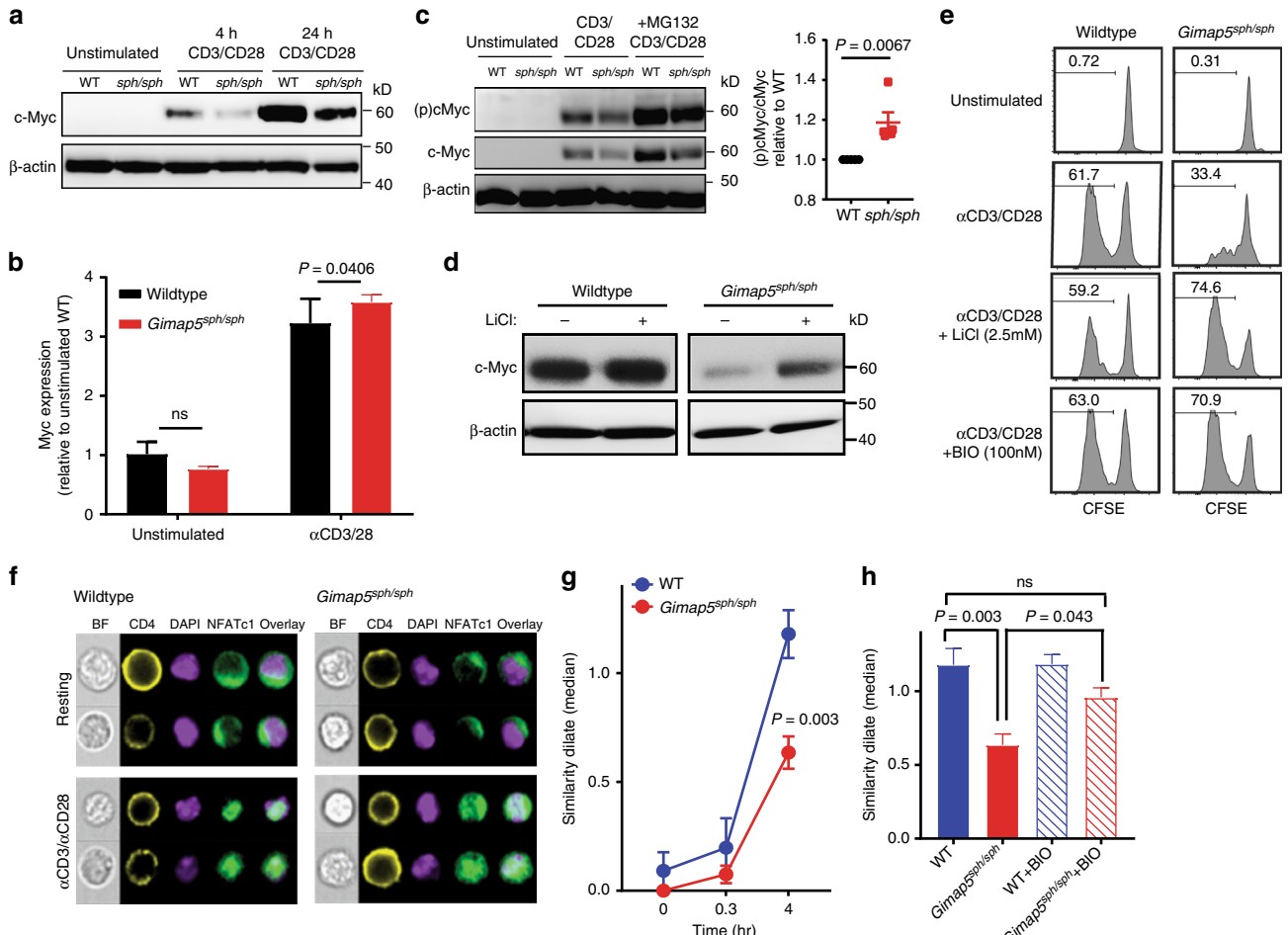

**Fig. 2** Impaired CD4[+] T cell proliferation is associated with increased GSK3 activity. **a** Immunoblot analysis of c-Myc expression in total lysates of CD4[+] T cells from WT and *Gimap5[sph/sph]* mice stimulated with αCD3/αCD28 or during resting conditions. **b** *Myc* mRNA levels in resting and αCD3/αCD28-activated (24 h) CD4[+] T cells. Data represent mean expression ± SD relative to unstimulated WT cells (*n* = 6). **c** Phosphorylation c-Myc (T58) after 24 h of αCD3/αCD28 stimulation. Proteasomal inhibitor MG132 was added after 20 h of stimulation. Ratio of p-c-Myc (T58) to total c-Myc in MG132-treated *Gimap5[sph/sph]* CD4[+] T cells relative to WT (*n* = 5). **d** C-Myc expression in WT and *Gimap5[sph/sph]* CD4[+] T cells stimulated for 24 h with αCD3/αCD28 ± 2.5 mM LiCl. **e** Proliferation of WT and *Gimap5[sph/sph]* CD4[+] T cells in the presence/absence of αCD3/αCD28 and/or GSK3 inhibitors BIO (100 nM) or LiCl (2.5 mM) as measured by CFSE dilution after 3 days. Experiments were repeated five times and representative plots from a single experiment are shown. **f** Representative images of NFATc1 localization in resting and αCD3/αCD28-activated WT and *Gimap5[sph/sph]* CD4[+] T cells. Bright detail similarity quantification of NFATc1 nuclear localization upon stimulation (**g**) and in the presence/absence of BIO (**h**) for 4 h (*n* = 6). Graphs depict mean values ± SEM. ImageStream data represent average values of >500 CD4[+] T cells per sample; all experiments were performed at least three times. Statistical significance is determined by Student's two-tailed test. BF bright field

specific activation of T cells, GSK3β activity is inhibited[19–22,37], thereby facilitating T cell activation and proliferation. GSK3 phosphorylates a variety of substrates, regulating protein ubiquitination/degradation and/or their activity[23]. Among the substrates targeted by GSK3 are c-Myc, Mcl-1, NFATc1, and β-catenin[24,38–41]. Elevated GSK3 activity in stimulated *Gimap5[sph/sph]* CD4[+] T cells would account for the reduced protein levels of Mcl-1 (Supplementary Figure 2E), which is targeted for proteasomal degradation upon phosphorylation by GSK3[39,40]. Moreover, αCD3/αCD28-stimulated *Gimap5[sph/sph]* CD4[+] T cells also showed a slight reduction in β-catenin protein expression, but normal *β-catenin* mRNA levels (Supplementary Figure 4A,B), consistent with an elevated GSK3 activity.

To further investigate whether GSK3 activity was elevated in the absence of Gimap5, we stimulated WT and *Gimap5[sph/sph]* CD4[+] T cells with αCD3/αCD28 for 24 h. During the final 4 h, proteasomal degradation was blocked with the inhibitor MG132 to allow the evaluation of GSK3 phosphorylation of c-Myc at

Thr58. Strikingly, *Gimap5[sph/sph]* CD4[+] T cells had a higher proportion of c-Myc phosphorylation than WT CD4[+] T cells (Fig. 2c). Moreover, the GSK3-inhibitor LiCl increased c-Myc and β-catenin accumulation in *Gimap5[sph/sph]* CD4[+] T cells following TCR stimulation (Fig. 2d and Supplementary Figure 4C). We thus hypothesized that impaired suppression of GSK3 activity compromises productive T cell proliferation in *Gimap5[sph/sph]* mice. To test this, CD4[+] T cells from WT and *Gimap5[sph/sph]* mice were stimulated with αCD3/αCD28 in the presence/absence of GSK3-inhibitors, LiCl or 6-bromoindirubin-3′-oxime (BIO). Strikingly, in the presence of GSK3-inhibitors, CD4[+] T cells from *Gimap5[sph/sph]* mice completely regained their proliferative capacity in vitro and showed comparable survival to WT CD4[+] T cells (Fig. 2e).

Finally, we assessed nuclear translocation of NFATc1, a TF necessary for productive T cell activation (Supplementary Figure 4D) and regulated by GSK3α/β[42,43]. Unlike c-Myc and β-catenin, however, GSK3 phosphorylation of NFATc1 regulates

its localization rather than stability[38,41]. To test if activation of NFATc1 was impaired in the absence of Gimap5, we stimulated WT and *Gimap5^{sph/sph}* CD4+ T cells with αCD3/αCD28 and examined NFATc1 nuclear localization by ImageStream analysis using the similarity algorithm (see Methods section). Notably, while expression of NFATc1 is normal (Supplementary

Figure 4E), its nuclear translocation is significantly impaired in stimulated *Gimap5^{sph/sph}* CD4+ T cells, and can be restored with BIO treatment (Fig. 2f–h). Furthermore, accumulation of intracellular calcium, a key step in calcineurin activation and NFATc1 translocation, was normal in *Gimap5^{sph/sph}* CD4+ T cells stimulated with either αCD3/αCD28 or ionomycin

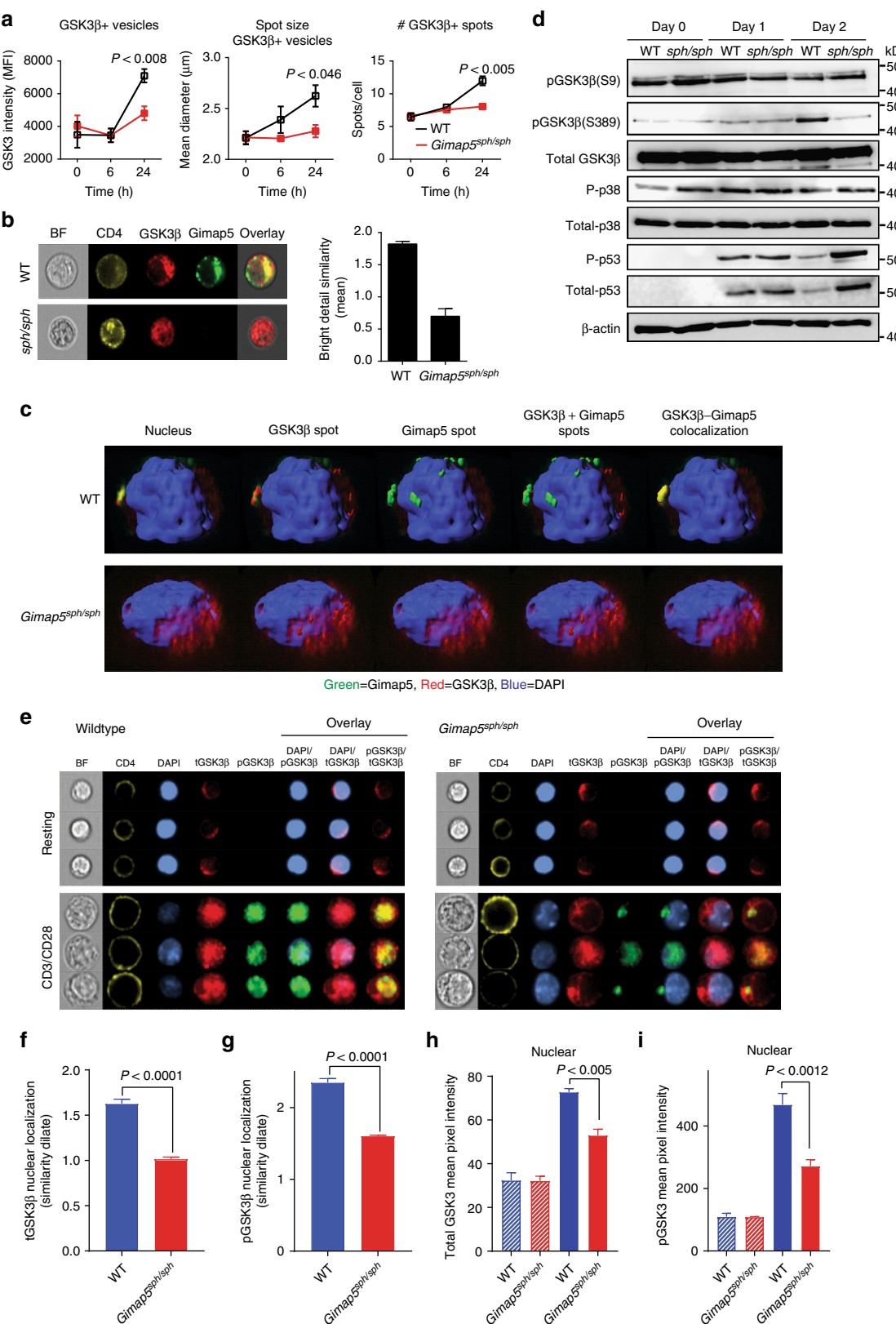

**Fig. 3** Loss of Gimap5 results in impaired inactivation of GSK3β. **a** Association of GSK3β with vesicles in WT and Gimap5[sph/sph] CD4[+] T cells as exemplified by GSK3β intensity, vesicle number, and vesicle size of GSK3β spots 6 and 24 h after αCD3/αCD28 activation. **b** Bright detail similarity analysis (ImageStream) of activated CD4[+]T cells from WT and Gimap5[sph/sph] mice show colocalization between Gimap5 and GSK3β. **c** Representative Z-stacks of WT and Gimap5[sph/sph] CD4[+] T cells stimulated for 24 h with αCD3/αCD28. **d** Immunoblot analysis of WT and Gimap5[sph/sph] CD4[+] T cells stimulated with αCD3/αCD28 depicting total and phosphorylated protein levels of GSK3β (P-Ser9 and P-Ser389), p38, and p53. **e–i** Localization of total and phospho-GSK3β (Ser389) in WT or Gimap5[sph/sph] CD4[+] T cells after 2 days stimulation with αCD3/αCD28 (n = 3). **f** Nuclear localization and **h** expression of total GSK3β. **g** Nuclear localization and **i** expression of p-GSK3β (Ser389). Hatched bars represent resting and solid bars represent CD3/CD28-activated CD4[+] T cells. Graphs depict mean values ± SD. ImageStream data represent average values of >500 CD4[+] T cells per sample. All experiments were performed at least three times. Statistical significance is determined by Student's two-tailed test. BF bright field

(Supplementary Figure 4F). Together these data suggest that the absence of Gimap5 causes impaired inactivation of GSK3, ultimately resulting in a failure to accumulate TFs and nuclear localization that are critically required for CD4[+] T cell survival/proliferation.

**Loss of Gimap5 results in impaired sequestration of GSK3β.** As mentioned, antigen-specific activation of T cells requires inactivation of GSK3 activity[22]; we therefore sought to determine how inactivation of GSK3β in Gimap5[sph/sph] CD4[+] T cells was impaired. The exact mechanism by which inhibition of GSK3β activity is regulated is not well defined. However, a variety of mechanisms have been proposed, including phosphorylation at residues Ser9[26] and Ser389[27] as well as changes in localization—i.e., physical sequestration of GSK3β in MVB[25]. Given that Gimap5 expression is observed in lysosomes and MVB[18], we first tested whether vesicular sequestration of GSK3β is affected. Specifically, we quantified vesicular GSK3β intensity, the number of vesicles, and the size of GSK3β-associating vesicles in WT and Gimap5[sph/sph] CD4[+] T cells at 0, 6, and 24 h after CD4[+] T cell stimulation. Notably, GSK3β became associated with punctate vesicles in WT cells and GSK3β intensity, vesicle number, and vesicle size were markedly decreased in Gimap5[sph/sph] CD4[+] T cells after 24-h stimulation (Fig. 3a), indicating that physical segregation of GSK3β from cytosolic target proteins[25,44,45] may provide a mechanism of inhibition. Next, we sought to identify if GSK3β[+] vesicles associate with Gimap5 expression following CD4[+] T cell activation. Interestingly, ImageStream analysis of resting/activated WT and Gimap5[sph/sph] CD4[+] T cells indeed showed colocalization between Gimap5 and GSK3β in a subset of punctate cytosolic vesicles of activated WT CD4[+] T cells (Fig. 3b) with ~54% of Gimap5[+] vesicles also positive for GSK3β. Given that Gimap5 is expressed in lysosomes (Lamp1[+]), but also Lamp1[neg] vesicles (Supplementary Figure 5B), we sought to further characterize Gimap5/GSK3β double positive (DP) vesicles using various endosomal/lysosomal markers. Interestingly, our studies showed limited association between Rab5, Rab7, or Lamp1 expression and Gimap5/GSK3β DP vesicles (Supplementary Figure 5A–E). To confirm colocalized expression of Gimap5 and GSK3β in vesicles, we examined WT and Gimap5[sph/sph] CD4[+] T cells by high-resolution, three-dimensional confocal microscopy, enabling us to visualize Gimap5-GSK3β colocalization in the axial as well as xy planes. Consistent with the ImageStream analysis, GSK3β colocalized with Gimap5 in stimulated WT CD4[+] T cells in punctate vesicles (Fig. 3c).

GSK3β vesicular sequestration has predominantly been described in the context of Wnt signaling[25]. To test the potential involvement of Wnt signaling, we incubated WT CD4[+] T cells either directly with Wnt3a or examined blocking of Wnt secretion by incubating activated CD4[+] T cells in the presence/absence of IWP-2 (a Wnt secretion inhibitor[46,47]). Subsequently, we assessed GSK3β sequestration using ImageStream and determined the proliferation capacity of WT CD4[+] T cells. Neither Wnt3a nor IWP-2 affected the vesicular sequestration in

WT CD4[+] T cells or impacted T cell proliferation (Supplementary Figure 6A–D). Moreover, no significant differences in Wnt3A mRNA expression were observed between WT and Gimap5[sph/sph] CD4[+] T cells at resting or activated conditions (Supplementary Figure 6E). Together, our data indicate that following activation of CD4[+] T cells, GSK3β is sequestered in vesicles and that this vesicular accumulation of GSK3β is markedly impaired in Gimap5-deficient CD4[+] T cells following activation.

**Impaired GSK3β phosphorylation and increased DNA damage.** Phosphorylation at residues Ser9[26] and Ser389[27] has been proposed as a mechanism of GSK3β inhibition. Phosphorylation of GSK3β at Ser9 is mediated by AKT (and possibly other kinases) following TCR signaling[48], while inhibition of GSK3β by phosphorylation of Ser389 is linked with lymphocyte fitness in the context of DNA DSB evident either during V(D)J recombination in thymic T cells or in B cells undergoing immunoglobulin CSR[27]. Furthermore, reports have shown that pharmacological inhibition and shRNA knockdown of GSK3β promote the repair of DSB, suggesting a role for GSK3β inhibition during the DNA damage response[28]. We thus next considered that GSK3β inhibition by phosphorylation—i.e., P-Ser9 or P-Ser389—during T cell activation was impaired in mutant cells. Interestingly, analysis of αCD3/αCD28-stimulated WT and Gimap5[sph/sph] CD4[+] T cells showed normal phosphorylation of Ser9 within the first 30 min (Supplementary Figure 7A,D), 24 h (Supplementary Figure 7A), and after 2 days (Fig. 3d). In contrast, phosphorylation of GSK3β at position Ser389 (primarily observed 2 days after T cell activation) was impaired in Gimap5[sph/sph] T cells (Fig. 3d). Phosphorylation of Ser389 is thought to be mediated by p38 MAPK[27]. However, we observed no differences in the activation of p38 MAPK between WT and Gimap5[sph/sph] CD4[+] T cells (Fig. 3d, Supplementary Figures 7C and 3B). To further define whether the reduced phosphorylation of GSK3β at position Ser389 was a result of an overall reduction in GSK3β nuclear expression, we performed ImageStream studies to define the level of both total GSK3β and P-Ser389 GSK3β in the nucleus. Interestingly, both total and phosphorylated GSK3β levels were markedly reduced in the nucleus of activated Gimap5[sph/sph] CD4[+] T cells (day 2) (Fig. 3e–i), while some P-Ser389 GSK3β could be observed in Gimap5[sph/sph] CD4[+] T cells predominantly in punctate vesicles outside of the nucleus (Fig. 3e), revealing an impaired distribution of P-Ser389 GSK3β in activated Gimap5[sph/sph] CD4[+] T cells.

Importantly, the reduction of P-Ser389 was associated with a marked increase in DNA damage (γH2AX expression; Fig. 4a) in Gimap5[sph/sph] CD4[+] T cells undergoing cell cycle, leading to increased p53 expression and phosphorylation (Fig. 3d), and ultimately reduced cell survival (Fig. 4b and Supplementary Figure 7G). Notably, Gimap5[sph/sph] CD4[+] T cells with DNA damage (γH2AX[+]) were not actively undergoing apoptosis (Supplementary Figure 7E,F). Importantly, the significant increase in DNA damage observed in Gimap5[sph/sph] CD4[+] T cells was reduced in the presence of GSK3 inhibitors (Fig. 4c, d), confirming the critical function of GSK3 inactivation in

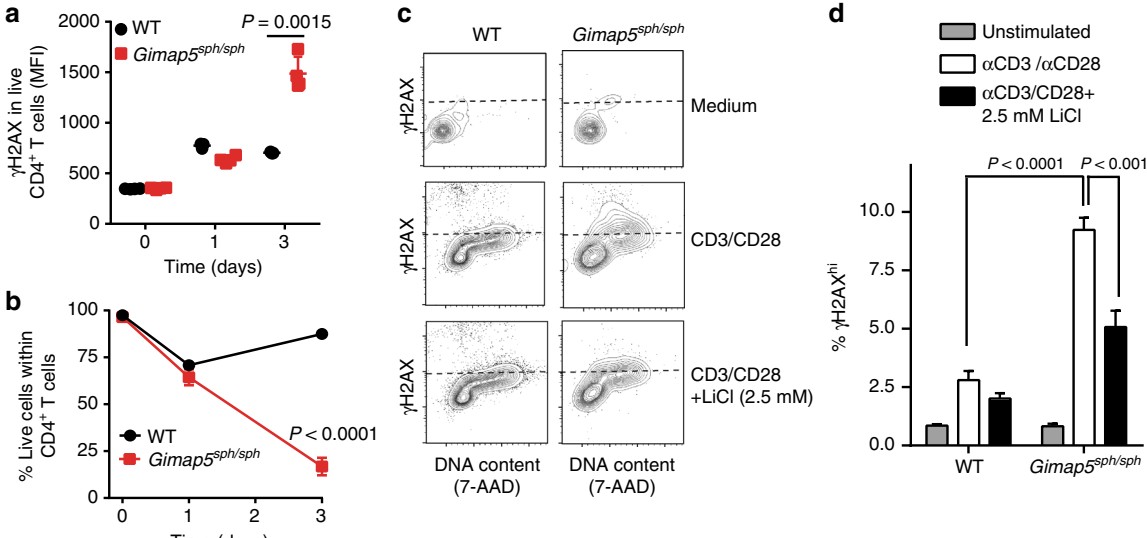

**Fig. 4** Loss of Gimap5 causes increased DNA damage in activated CD4⁺ T cells. **a** γH2AX expression in live WT or *Gimap5^{sph/sph}* CD4⁺ T cells following αCD3/αCD28 stimulation. **b** Survival as determined by viability stain of WT or *Gimap5^{sph/sph}* CD4⁺ T cells after αCD3/αCD28 stimulation (*n* = 4). Effect of lithium on γH2AX expression in WT or *Gimap5^{sph/sph}* CD4⁺ T cells, after 3 days stimulation with αCD3/αCD28. Plots depict live cells (**c, d**), while bar graphs (**d**) represent mean values ± SD (*n* = 6). All experiments were performed at least three times. Statistical significance is determined by Student's two-tailed test

limiting DNA damage and promoting productive T cell proliferation.

**Pharmacological targeting of GSK3 prevents immune pathology.** We next tested the potential of GSK3 inhibitors to maintain T cell survival and prevent colitis/liver pathology in vivo. We treated WT and *Gimap5^{sph/sph}* mice with LiCl (150 mg/kg in drinking water, ad libitum) starting at 3 weeks of age until 8 weeks of age, by which time severe colitis can be observed. LiCl treatment of *Gimap5^{sph/sph}* mice effectively maintained the CD4⁺ T cell and B cell populations in *Gimap5^{sph/sph}* mice (Fig. 5a, b), while the proportion of CD4⁺ T cells displaying the memory-like phenotype (CD44^{hi};CD62^{lo}) was reduced compared to untreated *Gimap5^{sph/sph}* mice (Fig. 5c).

Notably, while the Treg cell frequency within the CD4⁺ T cell population of *Gimap5^{sph/sph}* mice is not affected (Supplementary Figure 8C), their progressive loss of suppressive function is thought to be a key factor in driving colitis in *Gimap5^{sph/sph}* mice[5]. We thus next asked if pharmacological inhibition of GSK3 could prevent this defect. Specifically, we purified regulatory T cells from the spleens of vehicle- and LiCl-treated WT and *Gimap5^{sph/sph}* mice and cocultured them with αCD3-stimulated WT CD8⁺ T cells. Strikingly, while Treg cells from vehicle-treated *Gimap5^{sph/sph}* mice were incapable of suppressing CD8⁺ T cell proliferation in vitro, Treg cells from LiCl-treated *Gimap5^{sph/sph}* mice demonstrated normal suppressive capacity (Fig. 5d). Importantly, LiCl treatment completely prevented colitis and significantly reduced crypt loss and leukocyte infiltration (Fig. 5e, f and Supplementary Figure 8A). Finally, LiCl treatment corrected the regenerative liver phenotype with limited hemorrhage and hematopoietic stem cell presence observed in the liver of LiCl-treated *Gimap5^{sph/sph}* mice (Fig. 5g and Supplementary Figure 8B). Overall, these data provide a causal link between GSK3 activity and reduced T cells and B cell survival in *Gimap5^{sph/sph}* mice and the development of regenerative liver disease and colitis.

**Genetic targeting of GSK3β in CD4⁺ T cells prevents colitis.** While we observed comparable results in rescuing effects of loss of Gimap5 with two mechanistically distinct inhibitors of GSK3,

each of these molecules has other effects on cellular processes and proteins. GSK3 consists of two ubiquitously expressed paralogs, GSK3α and GSK3β; to assess selectivity and importance of GSK3β specifically in the CD4⁺ T cell-mediated pathology, we next examined whether genetic ablation of GSK3β in CD4⁺ T cells was sufficient to prevent loss of CD4⁺ T cells and immunopathology in *Gimap5^{sph/sph}* mice. To test this hypothesis, we generated WT- or *Gimap5^{sph/sph}*-; *Cd4-cre/ert2*; *Gsk3b^{fl/fl}* mice[49]—allowing for the conditional ablation of GSK3β specifically in CD4⁺ T cells upon treatment with tamoxifen. WT and *Gimap5^{sph/sph}* on a *Gsk3b^{fl/fl}* or *Gsk3b^{WT/WT}* background were treated with tamoxifen starting at 3 weeks of age. At 8 weeks of age, mice were characterized for CD4⁺ T cell survival and their phenotype. Tamoxifen treatment resulted in an effective ablation of GSK3β expression in CD4⁺ T cells (Supplementary Figure 9A) and increased survival of CD4⁺ T (but not B) cells in the periphery (Fig. 6a and Supplementary Figure 9B,C). Moreover, in the absence of GSK3β, *Gimap5^{sph/sph}* CD4⁺ T cells maintained a naïve phenotype comparable to WT peripheral CD4⁺ T cells (Fig. 6b). Importantly, conditional deletion of *Gsk3β* in CD4⁺ T cells was sufficient to prevent the development of colitis (Fig. 6c, d). In contrast, no effect on the liver damage with a similar presence of hematopoietic foci observed in tamoxifen-treated *Gimap5^{sph/sph}*; *Gsk3b^{fl/fl}* mice was observed (Supplementary Figure 9D). These findings confirm the critical function of GSK3β expression/activity in CD4⁺ T cell survival and T cell-mediated gut pathology in *Gimap5^{sph/sph}* mice.

**Human GIMAP5 LOF mutation results in T cell deficiencies.** During the course of our studies, we identified a 16-year-old patient who was referred to our institution for the evaluation of an immune deficiency. The patient presented with splenomegaly, lymphopenia with low CD4⁺ and CD8⁺ T cells as well as NK cells, and was found to have decreased frequency of naïve CD4⁺ T cells and increased CD4⁺ T-effector memory cells (see Supplementary Note 1 for patient description). Using whole-exome sequencing, the patient was found to be homozygous for a rare non-synonymous SNP (rs72650695, position 7:150742750) in the coding region of *GIMAP5* causing a Leu → Pro amino acid change

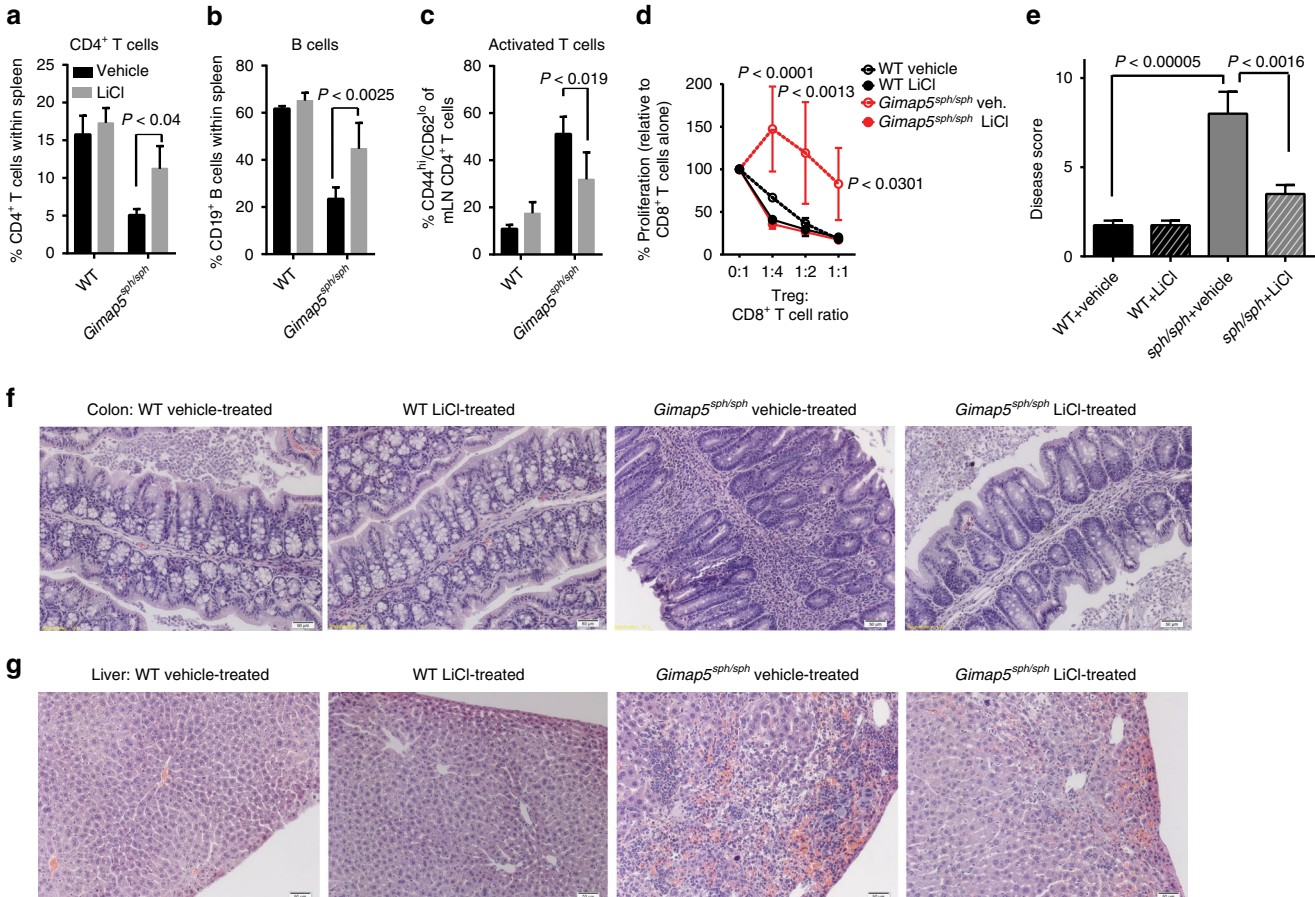

**Fig. 5** GSK3 inhibition improves lymphocyte survival and prevents immunopathology. LiCl treatment of *Gimap5^sph/sph* mice in vivo starting at 3 weeks of age and analyzed at 7–8 weeks of age, rescues CD4^+ T cell (**a**) and B cell (**b**) survival. **c** Reduced frequency of CD4^+ T cells undergoing lymphopenia-induced proliferation (CD44^hi;CD62^lo) upon LiCl treatment of *Gimap5^sph/sph* mice in vivo. **d** Suppressive capacity of regulatory T cells (CD4^+CD25^hi) isolated from vehicle or LiCl-treated WT and *Gimap5^sph/sph* mice. Data represents mean ± SD and is representative to two independent experiments (*n* = 3). Reduced colitis as defined by disease score (**e**) and based on colon histology (**f**) in 7–8-week-old *Gimap5^sph/sph* mice following LiCl treatment (3.5 weeks) in vivo. **g** Reduced liver pathology in *Gimap5^sph/sph* treated with LiCl. Data represent mean values ± SD from 6 mice per group at 7–8 weeks of age; histology is a representative depiction of disease severity. Scale bar is 50 μm. Statistical significance is determined by ANOVA followed by Sidak's multiple comparisons test

at residue 204 (Fig. 7a). Immunoblot analysis showed that the missense mutation resulted in undetectable protein expression of both GIMAP5 isoforms. (Fig. 7a). The rs72650695 SNP is a rare variant (minor allele frequency ~0.002076 in ExAC database) and both parents were found to be heterozygous carriers. Given that the lymphopenia and overall clinical profile were strikingly similar to Gimap5-deficient mice, and no other mutations in candidate genes were identified, we considered that the immune deficiency was caused by a LOF mutation in *GIMAP5*.

To define the T cell deficiency in the *GIMAP5^−/−* patient, we compared T cell proliferation from the patient with his heterozygous mother. Similar to CD4^+ T cell proliferation in Gimap5-deficient mice, a marked reduction in T cell proliferation was observed that could be rescued in the presence of LiCl (Fig. 7b). After expansion of T cells in vitro using phytohemagglutinin (PHA) followed by incubation with IL-2, rested T cells were examined for c-Myc expression. Immunoblot analysis showed a marked reduction of c-Myc levels in both resting (Fig. 7c) and αCD3/αCD28 re-stimulated patient cells as compared to the heterozygous mother (Fig. 7d). Importantly, similar to T cell proliferation, c-Myc levels could be restored upon treatment of cells with LiCl, suggesting that in human *GIMAP5*

^−/− T cells, GSK3 inhibitors offer therapeutic potential to correct the immunodeficiency and prevent immune pathology.

We next assessed colocalization of GIMAP5 and GSK3β in healthy control cells activated with αCD3/αCD28. Similar to previous studies, ImageStream analysis of primary CD4^+ T cells from healthy individuals showed that GIMAP5 was selectively expressed in vesicles, including but not limited to lysosomal vesicles (Fig. 8a, b). Moreover, we observed colocalization between GIMAP5 and GSK3β that was significantly increased 1–2 days after CD4^+ T cell activation with ~66% of GIMAP5^+ spots in CD4^+ T cells positive for GSK3β (Fig. 8a, c, d). Similar to murine CD4^+ T cells, accumulation of GSK3β was predominantly observed in GIMAP5^+ vesicles that were CD107 negative (Fig. 8c, d). Furthermore, vesicular association of GSK3β in *GIMAP5^−/−* CD4^+ T cells restimulated with αCD3/αCD28 after expansion was markedly reduced as determined by number of GSK3β spots, GSK3β spot size, and overall GSK3β intensity within spots (Fig. 8e–h). Notably, overall GSK3β expression and vesicular localization was also enhanced in CD4^+ T cells that had first been expanded compared to primary CD4^+ T cells both before and after αCD3/αCD28 stimulation (Fig. 8a, g, and h). We next determined whether the activation of *GIMAP5^−/−* CD4^+ T cells

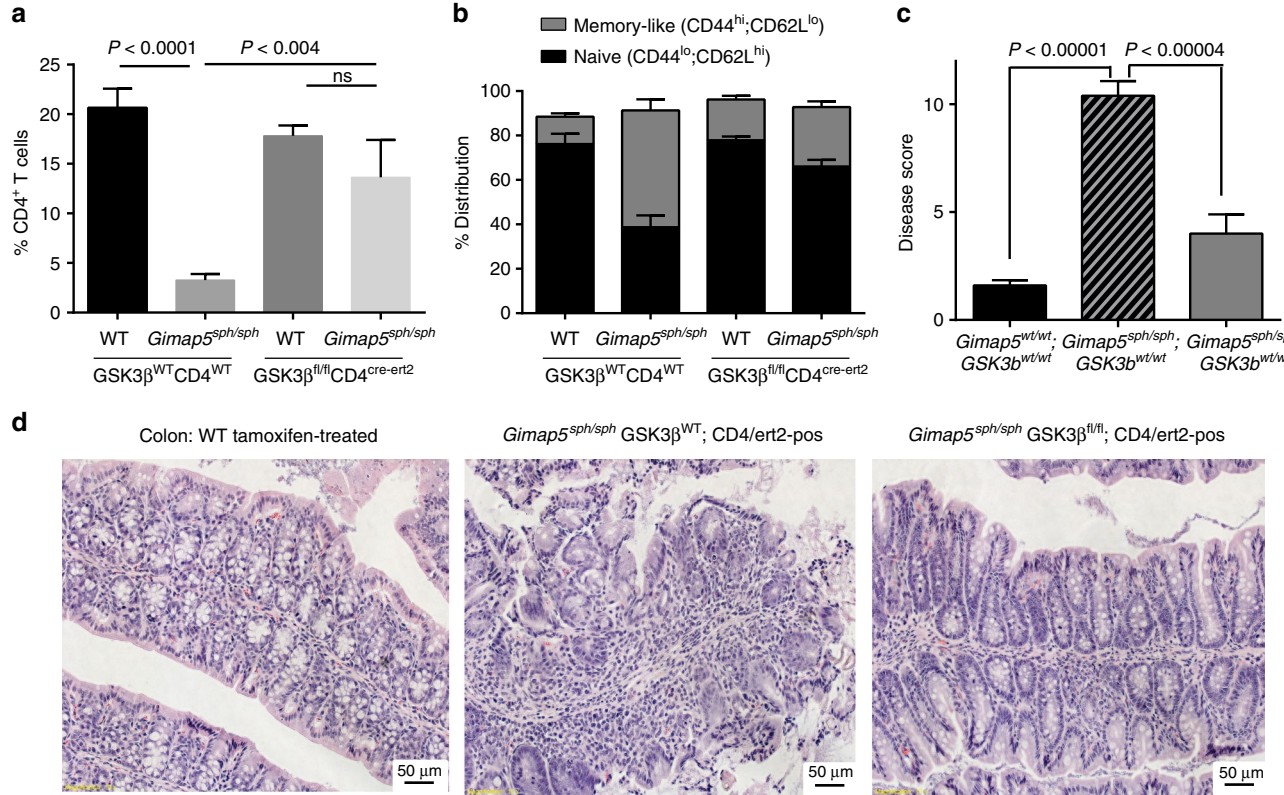

**Fig. 6** GSK3β deletion in CD4⁺ T cells improves CD4⁺ T cell survival and prevents colitis. **a** Tamoxifen treatment of *Gimap5^sph/sph*; *Gsk3b^fl/fl*; *Cd4cre-ert2* mice starting at 3 weeks of age selectively rescues splenic CD4⁺ T cell survival, **b** while maintaining overall CD4⁺ T cell quiescence. Reduced colitis in *Gimap5^sph/sph*; *Gsk3b^fl/fl*; *Cd4cre-ert2* mice treated with tamoxifen as determined by disease scores (**c**) based on histology (**d**). Data represent mean values ± SEM from at least 6 mice per group at 8 weeks of age; histology is a representative depiction of disease severity. Scale bar is 50 μm. Statistical significance is determined by ANOVA followed by Sidak's multiple comparisons test

was associated with increased DNA damage, by comparing the level of γH2AX in CD4⁺ T cells from control and patient cells. After 2 days of re-stimulation with αCD3/αCD28, a significant increase in intensity of γH2AX was observed in patient CD4⁺ T cells compared to control T cells upon activation (Fig. 8i, j). Together these data reveal a striking similarity between the molecular pathways in mouse and human T cells that are affected by loss of GIMAP5 function.

## Discussion

Here, we report a critical role for Gimap5 in inactivating GSK3β during CD4⁺ T cell activation that affects two key regulatory events required for T cell proliferation. In the absence of Gimap5, impaired inactivation of GSK3β through a reduced vesicular association results in a failure to accumulate or induce nuclear translocation of TFs necessary for productive T cell proliferation (Fig. 9). Moreover, when T cells are cycling (i.e., at day2), we observed an impaired nuclear accumulation of P-Ser389 GSK3β that was associated with increased DNA damage and a reduced lymphocyte fitness. Importantly, we show that pharmacological targeting of GSK3 or genetic deletion of *Gsk3b* corrects T lymphocyte survival and prevents severe early-onset colitis in Gimap5-deficient mice. Moreover, we describe a human patient with a GIMAP5 LOF mutation. The patient presents with an immunodeficiency strikingly similar to Gimap5-deficient mice, including the development of lymphopenia, reduction in TFs such as c-Myc, as well as increased DNA damage and reduced survival upon T cell activation. Notably, GSK3 inhibitors can

restore accumulation of c-Myc protein and improve T cell survival during activation in vitro.

GSK3β is constitutive active kinase in resting T cells and has a large number of reported targets (>100) that include important signaling components and TFs important for growth and survival[50]. These include but are not limited to c-Myc, β-catenin, and NFATc1—key TFs required for growth and activation of the transcriptional program in T cells[34,51,52]. In addition, our studies identify an important function for GSK3β in limiting DNA damage during CD4⁺ T cell proliferation. The latter could very well be mediated by direct effect of GSK3β on components such as p53, Mcl-1, but also Mdm2, a GSK3β target that is an important negative regulator of p53[53]. We tested the contribution of the pro-survival protein Mcl-1 as one potentially important survival factor, but our studies showed little rescue of CD4⁺ T cells in Bax/Bak-deficient mice, suggesting that a combination of pathways are likely underlying the failure of CD4⁺ T cells to survive and undergo proliferation. Thus, the relative contribution of these GSK3β targets in the overall Gimap5 phenotype is currently unclear and will be an important, but challenging question to address.

Until recently, AKT-mediated phosphorylation of the N-terminal tail of GSK3β was believed to be the primary mechanism of its inhibition. However, the generation of phospho-insensitive forms of GSK3 (GSK3α^S21A/GSK3β^S9A) and the characterization of GSK3β's active site have clearly demonstrated that GSK3 N-terminal phosphorylation neither completely inhibits activity nor is involved in every GSK3 signaling axis[44,54–56]. More recently, phosphorylation of the Ser389 site in GSK3β has

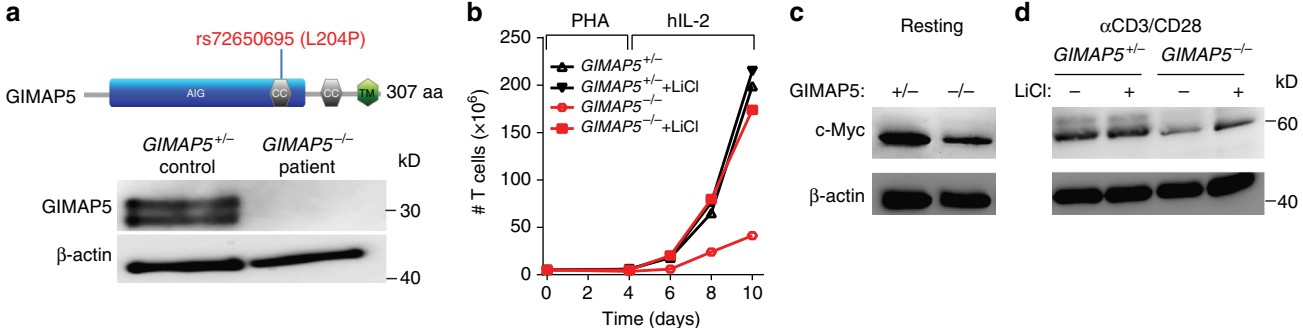

**Fig. 7** A human loss-of-function mutation in *GIMAP5* results in a similar T cell deficiency. **a** Whole-exome sequencing uncovered a homozygous variant (SNP: rs72650695), causing an L204P amino acid change in GIMAP5, resulting in complete loss of GIMAP5 protein expression in T cells. **b** Impaired expansion of CD3[+] T cells from *GIMAP5[−/−]* patient compared to heterozygous control cells following stimulation with PHA (4 days) and IL-2 (days 4–10). The proliferation capacity is restored in the presence of LiCl. Experiment is representative of three independent experiments from samples obtained several months apart. Immunoblot analysis of control and *GIMAP5[−/−]* CD3[+] T cells shows reduced c-Myc expression (**c**) at resting conditions and (**d**) after 2-day restimulation with αCD3/αCD28 ± 5 mM LiCl. Immunoblot was repeated twice with an identical outcome

been implicated as a key step in GSK3β inactivation and maintaining lymphocyte fitness during the DNA damage response to DSB[27]. Reports by Yang et al. indicate that reduction of GSK3β activity through inhibition or knockdown limits the accumulation of these DSB[28]. Other reports link GSK3 inhibition to increased cell survival upon induction of DNA damage by preventing its action on p53; active GSK3β binds to and promotes the actions of p53[57–59]. In addition, evidence suggests that the GSK3-mediated regulation of the transcriptional program (e.g. c-Myc, NFATc1) contributes to the optimal DNA repair response. For instance, consistent with our observations, inhibition of NFAT nuclear translocation impairs DNA repair upon UV irradiation[60,61]. Moreover, the report by Thornton et al. demonstrates the importance of phosphorylation of GSK3β at position Ser389 by using Ser389Ala knockin mice that have a reduced fitness of T cells and B cells undergoing V(D)J recombination and peripheral B cells undergoing activation[27]. However, P-Ser389 of GSK3β can also readily be observed following γ-radiation or upon treatment with doxorubicin[27,28], suggesting a broader context for this mechanism of GSK3β inactivation in limiting DNA damage. While the study by Thornton et al. suggests limited P-Ser389 after 18 h of stimulation in peripheral CD4[+] T cells[27], our studies show significant Ser389 phosphorylation at 48 h during the peak of the proliferation-associated DNA damage response in WT CD4[+] T cells—phosphorylation that is notably reduced in Gimap5-deficient T cells. Currently, it is unclear why Gimap5-deficient CD4[+] T cells fail to phosphorylate Ser389—a process thought to involve activation of Ataxia telangiectasia mutated (ATM) and phosphorylation by p38. Our data suggest phosphorylation of p38 is unaffected in Gimap5-deficient T cells, indicating that an alternative pathway is involved. Notably, at 24 h of activation, we did observe a marked decrease in the association of GSK3β with cytosolic vesicles—a mechanism proposed for GSK3 inhibition through the Wnt signaling pathway. This was followed by an overall reduced P-Ser389 and nuclear translocation of GSK3β, while remnant P-Ser389 GSK3β could be observed in punctate vesicles outside of the nucleus of *Gimap5[sph/sph]* CD4[+] T cells after 2 days stimulation. These observations suggest that Gimap5 facilitates the subcellular localization of GSK3β and imply that the nuclear translocation of GSK3β is required for a productive T cell response.

While dysregulation of GSK3 may be the most striking phenotype observed in the absence of Gimap5, it may not be the only effect of Gimap5 deficiency. Crystallographic studies revealed that Gimap proteins manifest a nucleotide coordination and dimerization mode similar to dynamin GTPase—a component essential for the scission and fusion of cellular vesicular compartments such as endosomes[62]. Thus, Gimap5's homology to dynamin raises the possibility that it may be more broadly involved in vesicular transport, rather than restricted solely to the sequestration of GSK3β. Interestingly, while Gimap5 is expressed in lysosomes and MVB, our data demonstrates that the increased vesicular association of GSK3β following T cell activation occurs predominantly in Lamp1/CD107 negative vesicles, suggesting lysosomal-independent functions for Gimap5.

A study suggested that Gimap5 can interact with Bcl2 family members to regulate apoptotic pathways[15]; however, our studies show that the loss of CD4[+] T cells could not be rescued when *Gimap5[sph/sph]* mice were crossed to the apoptosis-resistant Bim-deficient or Bax/Bak-deficient backgrounds. Nonetheless, the Bcl2 family member Mcl-1 is targeted for degradation by GSK3-mediated phosphorylation, allowing the release of pro-apoptotic binding partners. A study by Chen et al. suggested that loss of Gimap5 was associated with enhanced Mcl-1 degradation in hematopoietic stem cells (HSC) ultimately leading to compromised mitochondrial integrity and an overall reduced survival of HSC[7]. We observe a similar reduction in Mcl-1 levels in stimulated CD4[+] T cells, although the in vivo relevancy is unclear. This reduction is consistent with impaired GSK3 inhibition, providing an explanation for the increased Mcl-1 degradation. Moreover, our data indicate that treatment of *Gimap5[sph/sph]* mice with GSK3 inhibitors in vivo prevents/corrects the development of hematopoietic foci in the liver. This correction was not observed when GSK3β was genetically ablated in CD4[+] T cells, potentially indicating that HSC were directly targeted by LiCl treatment. This is consistent with the observation that the liver phenotype still develops in *Gimap5[sph/sph]* mice on a Rag-deficient background, which lacks CD4[+] T cells[8].

It is feasible that GSK3 dysregulation has more subtle effects than activation-induced T cell death. We have previously shown that a subset of highly pathogenic memory-like CD4[+] T cells survives and drive colitis in the *Gimap5[sph/sph]* mouse. These remaining cells have a limited capacity to proliferate but robustly produce IFNγ and IL-17a. The accumulation of these cells and concurrent loss in number and function of regulatory T cells in Gimap5-deficient models results in an imbalance between Th17 and Treg cells that promotes the development of autoimmune conditions, including severe colitis (mice), aggravated EAE (LEW rats), and T1D (BB rats)[5,6,10,11,13]. Although the roles of Wnt signaling, GSK3, and β-catenin in CD4[+] T cell polarization and activity are controversial[19,20,63–67], dysregulation of GSK3 activity in CD4[+] T cells has been linked with skewing toward pathogenic

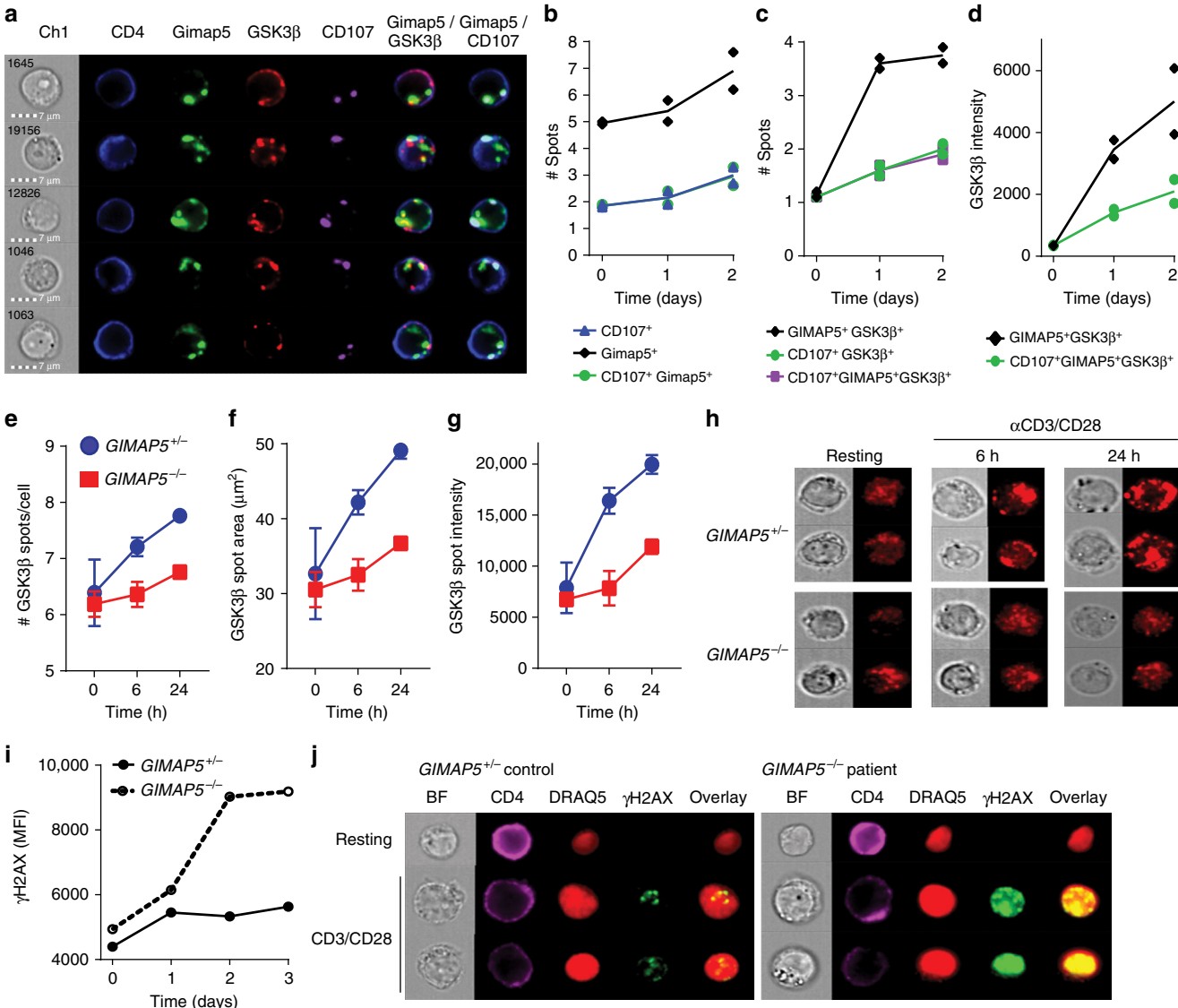

**Fig. 8** Human GIMAP5$^{-/-}$ T cells show impaired GSK3β sequestration and increased DNA damage. **a**–**d** Colocalization and spot analysis of GIMAP5$^+$, GSK3β$^+$, and CD107$^+$ vesicles in primary CD4$^+$ T cells isolated from healthy controls at a resting state (day 0) or after αCD3/αCD28 stimulation for 1–2 days (representative images from day 1 shown) using ImageStream analysis. **e**–**h** ImageStream analysis of GSK3β-specific vesicle association in control or GIMAP5$^{-/-}$ patient CD4$^+$ T cells restimulated with αCD3/αCD28 after primary expansion. Data depicts **e** GSK3β$^+$ spot number, **f** GSK3β intensity therein, and **g** spot area in live CD4$^+$ T cells at 0, 6, or 24 h of αCD3/αCD28-restimulation. Data represent mean values ± SD. **h** Representative images of GSK3β vesicular association in control and GIMAP5$^{-/-}$ CD4$^+$ T cells taken using a 60× objective. Analysis of DNA damage response (γH2AX) in control of GIMAP5$^{sph/sph}$ CD4$^+$ T cells after **i** 1–3 days or **j** 2 days restimulation with αCD3/αCD28. Data represents mean values of a single experiment performed in duplicate and repeated twice. ImageStream data represent average values of >500 CD4$^+$ T cells per experiment. BF bright field

Th17 cells. Studies by Beurel et al. have shown that upregulation of GSK3 promotes Th17 polarization, while its inhibition blocks this process[20]. In contrast, inhibition of GSK3 activity potentiates the polarization and suppressive capacity of Treg cells[63,66,68]. This provides a potential mechanism as to why Gimap5$^{sph/sph}$ Treg cells have a reduced suppressive capacity that can be restored by long-term LiCl treatment.

We show that a dysregulation of GSK3 in lymphocytes can lead to a primary immune deficiency, as observed in Gimap5$^{sph/sph}$ mice and the GIMAP5 LOF patient. Given its importance in maintaining CD4$^+$ T cell homeostasis and differentiation, we propose that even a minor dysregulation of GSK3 within the T cell compartment could result in abnormal polarization of CD4$^+$ T cells, leading to preferential differentiation of Th17 cells over regulatory T cells. Such a dysregulation may be occurring in

autoimmune diseases associated with pathogenic Th17 cells, such as Crohn's disease, T1D, and multiple sclerosis[69–71], or in Gimap5-associated diseases such as SLE, T1D, and allergic asthma[1–4]. In this case, selective inhibition of GSK3 activity offers a new promising therapeutic approach for treating patients with immunopathogenic T cell responses.

T cells depend on their ability to undergo clonal expansion for an efficient immune response during infection or to maintain immune homeostasis in the gut. Our studies reveal a key role for Gimap5 in inactivating GSK3β during CD4$^+$ T cell activation, a link that is critically required to maintain T cell fitness and allows for productive T cell proliferation. We propose that the Gimap5-mediated inactivation of GSK3β is an essential molecular mechanism to support productive CD4$^+$ T cell responses. Moreover, our studies point to a remarkable therapeutic potential

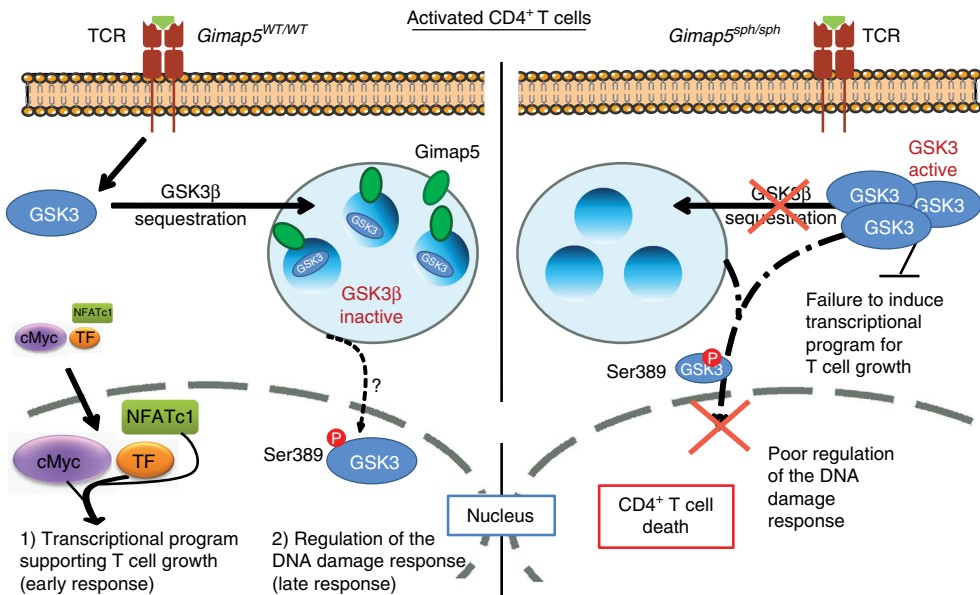

**Fig. 9** Gimap5 is a critical regulator of GSK3β during T cell activation. Gimap5 controls regulation of GSK3β in T cells through vesicular sequestration and affects both the (1) (early) transcriptional program required for T cell growth, and (2) the late stage nuclear accumulation of P-Ser389 GSK3β required for the DNA damage response during cycling. Gimap5-deficient CD4+ T cells, fail to inhibit GSK3β leading to a failed transcriptional program and increased DNA damage during T cells proliferation

for GSK3 inhibitors to improve CD4+ T cell survival/proliferation and prevent immunopathology. Thus far, GSK3 inhibitors have been used to treat a variety of diseases including Alzheimer's disease, mood disorders, cancer, and diabetes mellitus (for extensive reviews see[72,73]). Our current data reveal a new therapeutic application of GSK3 inhibitors specifically in the treatment of immunodeficient patients that have *GIMAP5* LOF mutations. These patients present a strikingly similar phenotype to Gimap5-deficient mice and suffer from recurrent (viral) infections most likely stemming from an overall lack of T cell fitness. We posit that GSK3-inhibitors will improve overall T cell survival and function and may prevent/correct immune-associated sequelae observed in these patients. In addition, therapeutic targeting of this pathway may be relevant for the treatment of patients with SNPs in *GIMAP5* linked to development of islet autoimmunity in T1D, SLE[1–3], or asthma[4].

## Methods

**Study design**. For the proposed experiments involving the studies of immune responses in vivo, both male and female mice were used. All strains of mice used were generated on a C57BL/6J background and confirmed by whole-genome SNP analysis and we anticipated limited genetic variation. To minimize confounding secondary factors arising from lymphopenia and other late-stage pathologies that develop in *Gimap5*[sph/sph] mice, CD4+ T cells from 3-week-old mice were used unless otherwise noted. At this age, *Gimap5*[sph/sph] mice have relatively normal numbers of CD4+ T cells with a normal frequency of naïve T cells that have a quiescent phenotype comparable to WT mice. Since we were interested in T cell proliferation/survival in WT vs. Gimap5-deficient cells, we expected to find strong differences in measurement (e.g., normal vs. absence of T cell survival/proliferation). In cases where the differences in the observed phenomena were clear and distinct, a group size of 6 mice was suitable to give statistically significant (i.e., $P < 0.05$) data. For these studies, we estimated that, with a sample size of 6, we have a 99% power to detect at least 25% reduction in the percentage of proliferating cells in stimulated *Gimap5*[sph/sph] cells with a significance level (alpha) of 0.05 (two-tailed).

**Mice and reagents**. All experiments were performed according to the US National Institutes of Health guidelines and were approved by the IACUC of The Cincinnati Children's Hospital. C57BL/6 J mice were obtained from Jackson. *Gimap5*[sph/sph] mice were generated as described[6] and bred in-house to generate WT or *Gimap5*[sph/sph];*Rag2*[−/−];OT-II, WT or *Gimap5*[sph/sph];Tg (*Cd4-cre/ert2*) 11Gnri/J; *Gsk3b*[fl/fl] mice in the vivarium of Cincinnati Children's Hospital. All mice were maintained under specific pathogen-free conditions. Purified α-mouse-CD3 (17A2)

and α-mouse-CD28 (37.51) antibodies (Biolegend) were used for murine T cell activation. For human T cell studies, purified αCD3 (OKT3) and αCD28 (CD28.2) antibodies (Biolegend) were used. 7-AAD was purchased from BD. Ovalbumin, LiCl, BIO, ionomycin, and phytohemagglutinin-L (PHA-L) were obtained from Sigma. Monoclonal antibody MAC421 was used as probe for Gimap5 as described[6]. GSK3β was probed with monoclonal antibody Clone 7/GSK-3b from BD. Information on all antibodies used can be found in Supplementary Table 1.

**T cell analyses**. To characterize lymphocyte populations ex vivo, lymphocytes from spleen and mesenteric lymph node (mLN) were isolated and stained with fluorochrome-conjugated antibodies for mouse CD4, CD8, CD19, B220, CD25, CD44, CD62L, and Foxp3. In all experiments, samples were stained with a fixable viability dye; analysis was restricted to live cells unless otherwise stated.

For all in vitro murine CD4+ T cell experiments, unless otherwise noted, Mojo-purified (Biolegend) CD4+ T cells were stimulated with plate-bound αCD3 (1 μg/mL) + soluble αCD28 (2 μg/mL). Cells were cultured in supplemented IMDM medium containing 10% FBS, 2% penicillin/streptomycin, 1% L-glutamine, and 50 μM β-mercaptoethanol (BME). T cell proliferation was quantified by incubating CD4+ T cells in 5 μM carboxyfluorescein succinimidyl ester (CFSE) in 0.2% FBS for 5 min. Cells were either left unstimulated or stimulated in the presence of GSK3 inhibitors LiCl (2.5 mM) or BIO (100 nM). After 3 days incubation, proliferation was evaluated by analyzing CFSE dilution by flow cytometry. In indicated experiments, 200 ng/mL Wnt3a, 2 μM IWP-2, or 200 ng/mL Cyclosporin A (CsA) were added. DNA damage was evaluated by γH2AX staining in conjunction with 7-AAD staining for cell cycle analysis. Early phosphorylation of GSK3β (S9), Akt (S473), and p38 (T180/Y182) was evaluated by stimulating rested CD4+ T cells with αCD3 (5 μg/mL)/αCD28 (2 μg/mL) for the indicated time point before staining for flow cytometry. Early phosphorylation of AKT, p38, ERK, and JNK, and the degradation of IκB was also evaluated in CD4+ T cells upon stimulation with phorbol 12-myristate 13-acetate (PMA) (50 ng/mL) and ionomycin (750 ng/mL). Ex vivo survival of CD4+ T cells was evaluated by culturing Mojo-purified peripheral CD4+ T cells or total thymocytes with 5 ng/mL IL-7 for 0–7 days. The number of surviving cells was determined by staining for CD4 and viability and counted on a flow cytometer using counting beads (Biolegend).

**Immunoblotting**. Protein lysates from human T cells were prepared according to standard methods from resting cells or cells stimulated with 5 μg/mL αCD3 + 2 μg/mL αCD28 ± 5 mM LiCl for 24 h. For mouse experiments, Mojo-isolated (Bio-Legend) CD4+ T cells were stimulated with αCD3 and αCD28 ± 2.5 mM LiCl for the indicated time periods prior to preparing protein lysates. In indicated experiments, 10 μM of proteasomal inhibitor MG132 was added 4 h prior to lysis of the cells. Lysates were separated using 10% Bis-Tris Gels, transferred to nitrocellulose, and immunoblotted with primary antibodies to GIMAP5, phospho-c-Myc (T58), c-Myc, pGSK3β (S389), pGSK3β (S9), total GSK3β, phospho-p53 (S15), total p53, phospho-p38 (T180/Y182), total p38, p-mTOR, total mTOR, p-p70 S6K (T389),

p70-S6K, β-catenin, and β-actin. All information on antibodies used can be found in Supplementary Table 1.

**AMNIS ImageStream flow cytometry.** For mouse localization studies, CD4$^+$ T cells were stimulated with αCD3 + αCD28. After 6, 24, or 48 h CD4$^+$ T cells were stained with antibodies to CD4 (GK1.5), Gimap5 (MAC421), GSK3β (BD 610202), pGSK3β (S389) (EMD Millipore 07–2275), DAPI, and a fixable viability dye. In indicated experiments, 200 ng/mL Wnt3a and 2 μM IWP-2 were added. For NFATc1 localizations studies, isolated CD4$^+$ T cells were stimulated indicated times. Cells were stained with antibodies for CD4 and NFATc1 (clone 7A6), DAPI, and a fixable viability dye. Live CD4$^+$ T cells were analyzed for NFATc1 and DAPI localization by delineating regions of positive signal (i.e., a mask). Nuclear translocation was measured by similarity dilate, which represents the log transformed Pearson's Correlation Coefficient and is a measure of the degree to which two images are linearly correlated within a masked region. In human GIMAP5-localization studies, CD4$^+$ T cells were isolated from the peripheral blood mononuclear cells (PBMCs) of healthy donors by MACS-purification (Miltenyi Biotec) and stimulated with αCD3 (5 μg/mL) and αCD28 (2 μg/mL). After 24 and 48 h, T cells were stained with antibodies to CD4, GIMAP5 (CST 14108), GSK3β (BD 610202), CD107b (Miltenyi Biotec), and a fixable viability dye. GSK3β vesicular localization in patient and control CD4$^+$ T cells was measured upon restimulation of resting T cells after IL-2 expansion. Colocalization, vesicular localization, and nuclear localization was evaluated using ImageStream Data Exploration and Analysis Software (IDEAS) 6.1 as described in detail[74]. Specifically, colocalization was quantified using the Bright Detail Similarity representing the log transformed Pearson's correlation coefficient of the localized bright spots with a radius of 3 pixels or less within the masked area in the two input images.

**Confocal microscopy.** To visualize the localization of GSK3β and Gimap5 at a higher resolution, CD4$^+$ T cells isolated from the spleen and lymph nodes of WT and Gimap5$^{sph/sph}$ mice were stimulated with αCD3/αCD28. After 24 h, cells were stained with antibodies for Gimap5 (MAC421) and GSK3β (BD61202). Cells were counterstained with DAPI (1 μg/mL) and mounted in Prolong Gold anti-fade reagent (Cell Signaling Technology). Samples were imaged on a Nikon A1 LUN-V inverted microscope using a 100× objective with oil immersion. For each imaged cell, Z-stacks were generated by taking images at a 0.125 μm step. To refine localization, images were deconvolved using the Landweber algorithm (15 iterations) in NIS Elements v4.5 (Nikon). Z-stacks were assembled and GSK3β and Gimap5 localization assessed in Imaris Image Analysis software v8.3 (Bitplane).

**Calcium flux.** Splenocytes isolated from individual WT or Gimap5$^{sph/sph}$ mice were stained with Indo-1 (2 μg/mL) at $1 \times 10^6$ cells/mL for 30 min at 37 °C. Cells were then stained for CD4 and rested for ≥1 h in cell loading medium (HBSS + 1 μM CaCl$_2$ + 1 μM MgCl$_2$ + 1% FBS) at room temperature. Five minutes prior to analysis, cells were warmed to 37 °C. To evaluate calcium flux, samples were acquired for ~30 s without stimulation and an additional 270 s after stimulation with αCD3/αCD28 (1.25 μg/mL) or ionomycin (800 ng/mL) for a total of 5 min.

**Real-time PCR.** Mojo-purified (Biolegend) CD4$^+$ T cells were rested or stimulated with αCD3/αCD28 for 24 h. Cells were lysed with TRIzol (Thermo Fisher Sci), mRNA isolated, and reverse transcription performed using a High-capacity cDNA Reverse Transcription Kit (Applied Biosystems). cDNAs were amplified with LightCycler 480 SYBR Green I Master (Roche) and quantified by Light-Cycler 480-II instrument (Roche). The following primer pairs were used: myc, forward: 5′-ATGCCCCTCAACGTGAACTTC-3′, reverse: 5′-GTCGCAGATGAAA-TAGGGCTG-3′; wnt3a, forward: 5′-CTCCTCTCGGATACCTCTTAGTG-3′, reverse: 5′-CCAAGGACCACCAGATCGG-3′; ctnnb1 (β-catenin), forward: 5′-ATGGAGCCGGACAGAAAAGC-3′, reverse: 5′-CTTGCCACTCAGGGAAGGA-3′; L32, forward: 5′-GAAACTGGCGGAAACCCA-3′, reverse: 5′-GGATCTGGCCCTTGAACCTT-3′. Expression of myc, wnt3a, and ctnnb1 was normalized to L32 and set relative to unstimulated WT samples.

**In vivo OVA administration.** The role of TCR signaling in Gimap5$^{sph/sph}$ CD4$^+$ T cell survival in vivo was evaluated by crossing Gimap5$^{sph/sph}$ mice with Rag2$^{-/-}$; OT-II mice. OVA (1 mg/mL) was administered to 10-week-old mice in drinking water ad libitum. After 5 weeks, T cell survival and Treg cell-induction were evaluated by flow cytometry.

**In vivo lithium treatment.** To evaluate if in vivo inhibition of GSK3 activity could prevent the development of Gimap5$^{sph/sph}$-associated pathologies, 3-week-old mice were administered 150 mg/kg LiCl in drinking water. After 4–5 weeks, liver damage, colitis, and lymphocyte populations were evaluated by histology and flow cytometry, respectively. Animals were assigned to treatment and vehicle groups randomly, with equal numbers of male and female mice assigned to each group. For these experiments, WT and Gimap5$^{sph/sph}$ genotypes were cohoused during treatment. Investigators were blinded to mouse genotype and treatment status during analysis.

**Genetic deletion of GSK3β.** GSK3β hyperactivation in Gimap5$^{sph/sph}$ CD4$^+$ T cells was evaluated by crossing Gimap5$^{sph/sph}$; Gsk3b$^{fl/fl}$ mice to Cd4cre-ert2 mice, allowing for the tamoxifen-inducible genetic deletion of Gsk3b in CD4$^+$ T cells. Tamoxifen was administered in food to 3-week-old mice (40 mg/kg body weight; Harlan Laboratories Teklad Diets). At 8 weeks, liver damage, colitis, and lymphocyte populations were evaluated by histology and flow cytometry, respectively.

**Histology.** Colon tissue was collected and immediately fixed in 10% buffered formalin solution overnight, followed by routine paraffin embedding. Hematoxylin and eosin staining were performed on 4 μm sections from the paraffin-embedded tissue blocks for conventional light microscopy analysis. Histological scoring was performed double-blind as described before[5,6]. Briefly, scoring parameters included quantitation of the area of distal colon involved, edema, erosion/ulceration of the epithelial monolayer, crypt loss/damage, and infiltration of immune cells into the mucosa. Severity for the area involved (erosion/ulceration and crypt loss) was graded on a scale from 0 (normal), 1 (0–10%), 2 (10–25%), 3 (25–50%), and 4 (>50%). Immune cell infiltration was scored as 0, absent; 1, weak; 2, moderate; and 3, severe. Total disease score was expressed as the mean of all combined scores per genotype.

**In vitro regulatory T cell suppression assay.** Treg cell suppression assays were performed as described using Treg cells isolated from the spleens of WT and Gimap5$^{sph/sph}$ mice treated with either LiCl or vehicle[5]. In brief, CD4$^+$ T cells were enriched by magnetic separation (Mojo, Biolegend) and stained for viability, CD4, and CD25. Live CD4$^+$CD25$^+$ regulatory T cells were isolated by FACS using a Beckman Coulter MoFlo XDP cell sorter. Sorted Treg cells were cocultured with the indicated ratios with $5 \times 10^4$ CTV-labeled CD8$^+$ T cells isolated from a naïve mouse. An additional $1 \times 10^5$ T cell depleted, gamma-irradiated (1500 rad) splenocytes were also cocultured as bystander cells. CD8$^+$ T cells were stimulated with 0.5 μg/mL αCD3; proliferation was assessed by CTV dilution after 3 days of coculture.

**Expansion of human T cells.** For all studies concerning human cells, informed consent was obtained and studies were approved by the CCHMC institutional IRB. PBMCs were isolated from whole blood of a GIMAP5-deficient patient and a healthy parent by Ficoll-Paque Plus density gradient centrifugation. Isolated PBMCs were cultured in RPMI, 10% FBS, 100 U/mL penicillin, 100 μg/mL streptomycin, 2 mM L-glutamine, and 10 mM HEPES and were stimulated with 5 μg/mL PHA-L ± 5 mM LiCl. After 4 days, T cells were selectively expanded in 150 U/mL rhIL-2 (Miltenyi) ± 5 mM LiCl for 8 days. Expanded T cells were rested for 2 days in the absence of IL-2 and LiCl before use in reactivation experiments. Prior to expansion, patient's PBMCs were stained for CD3, CD4, and CD8 and analyzed by flow cytometry for evaluation of circulating lymphocyte populations.

**Analysis of DNA damage response in human CD4$^+$ T cells.** Rested patient T cells were restimulated with 5 μg/mL αCD3 and 2 μg/mL αCD28. After 24, 48, and 72 h stimulation, cells were stained for viability, CD3, CD4, CD8, and γH2AX and analyzed by flow cytometry. At 24 and 48 h, cells were stained with α-CD4, α-γH2AX, DRAQ5, and a fixable viability dye. Analysis was performed on live CD4$^+$ T cells using ImageStream flow cytometry as detailed above.

**Statistical analysis.** All data were analyzed using GraphPad Prism4® software (GraphPad Software, San Diego, CA). For studies comparing T cells from C57BL/6 J, Gimap5$^{sph/sph}$; and/or Gsk3b$^{flox/flox}$ mice or involving chemical treatment of T cells, Student's two-tailed test or ANOVA followed by Sidak's multiple comparisons test for three or more groups were used. Data were considered statistically significant if $P$ values were <0.05. Data were normally distributed.

**Data availability.** The data that support the findings of this study are available from the corresponding author upon reasonable request.

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

## Acknowledgements

We thank G. Butcher for graciously providing the MAC421 αGimap5 antibody. The research was funded by the Crohn's & Colitis Foundation of America (#3793) and the NIH, NIDDK P30 DK078392 (Integrative Morphology Core of the Cincinnati Digestive Disease Research Core Center).

## Author contributions

A.P., M.E., K.L., and H.I.A. performed the experiments. J.R.W. and D.H. provided the mice. M.J. and H.S. provided the reagents and helped design the experiments. A.F., Z.K., and J.B. were involved in the patient differential diagnosis and patient treatment. A.P. and K.H. wrote the manuscript.

## Additional information

**Competing interests:** The authors declare no competing financial interests.

