## [Peer review file · Nature Communications]

Reviewers' comments:

Reviewer #1 (Remarks to the Author):

In their manuscript, Patterson et al identify a critical role for Gimap5 in GSK3 β phosphorylation that accounts for defects in the Gimap5-deficient mice including DNA damage responses, T cell loss and unopposed T cell activation. The manuscript is comprehensive including in vitro data and in vivo data in the Gimap5-deficient mice and identification and characterization of a human individual with a Gimap5 defect. Overall, the data are convincing and the findings are novel. The authors may want to consider the following points:

1. They convincingly demonstrate that GSK3 β and Gimap5 colocalize but it remains unclear how this facilitates phosphorylation. Do the authors think that Gimap5 functions as a scaffolding molecule and facilitates GSK3 β phosphorylation through p38. At least some discussion would be helpful.
2. More background information on how unopposed GSK3 β activity induces DNA damage and T cell loss would be helpful.
3. The authors do not describe b-catenin data of Figure 6d in the result section. Also it looks like b-catenin levels are not really different, contrary to the interpretation of the authors in discussion section (line 289-290).
4. In line 130, fig.S1G is incorrect. It should be fig.S1F.

Reviewer #2 (Remarks to the Author):

The Gimap GTPases are a set of septin/dynammin-related proteins which are expressed prominently in the haematopoietic lineages of mammals (and probably of other vertebrates), in particular in T lymphocytes. Mutations in rats and/or mice of two members of the gene family, Gimap5 and Gimap1, lead to substantial lymphopenias but the basis/bases of these detrimental effects are yet to be established. This paper addresses Gimap5 and proposes its involvement in the regulation of the activity of glycogen synthase kinase 3 β (GSK3 β). It proposes that Gimap5 is a negative regulator of GSK3 β activity and helps to constrain the degree to which this Ser/Thr kinase promotes the degradation of some factors crucial to aspects of T cell proliferation (notably c-Myc). It goes on to propose that in the absence of Gimap5 the inhibition of GSK3 β is impaired, with deleterious consequences for this proliferation and for cell survival, with increased levels of DNA damage.

The manuscript contains some compelling data in support of the authors' hypothesis. At the whole-animal level, the ameliorative effects on the sph/sph deficiency phenotypes of GSK3 β ablation and GSK3 β inhibitors are impressive. The narrative put together to develop a hypothesis for the potential molecular mechanism of Gimap5 function is very interesting. At the same time, however, the manuscript suffers from some substantial deficiencies in respect of (1) the quality and presentation of some of the data and (2) the writing of the manuscript, in particular the Results section, which suffers from a rather incoherent style and from inaccuracies in the proper description of the data (inaccurate scientific language). (The Discussion section is better written). The poor quality of presentation should have been evident to at least some of the long list of co-authors and I am surprised at the carelessness of submitting a paper to this journal without due attention.

Major points

1. The argument and data presented surround the effects of Gimap5 deficiency on events following conventional T cell activation via the TCR. What is not clear is how this relates to the primary aspect of the phenotype associated with Gimap5 deficiency, namely the in vivo T cell lymphopenia. Gimap5-deficient animals fail to maintain a population of naive, resting T cells. Part of this phenotype probably corresponds to a reduced thymic output of T cells and another part to a failure to maintain T cells in the resting condition (this point has been explicitly demonstrated in experiments on Gimap1-related lymphopenia). Neither of these stages corresponds obviously to

explicit antigen-driven T cell responses. Late SP thymocytes and resting T cells express notably high levels of Gimap5. The authors should address this issue.

2. The authors have conducted the majority of their studies using T lymphocytes from 3-wk old mice, when the mutant mice possess a phenotypically fairly normal T cell population, in contrast to later time points. The assumption that the T lymphocytes at this stage are 'normal', however, is not without risk. For instance, Gimap5-deficient SP thymocytes have a survival deficiency and this may extend into the cell stage they are studying. In the same vein, calcium responses of Gimap5-deficient SP thymocytes are reported to be down relative to WT (Ilangumaran et al. *Mol Immunol* 2009). A system in which Gimap5 could be electively deleted in lymphocytes would have been preferable.

3. Mcl-1. As a target of GSK3 β , Mcl-1 is of interest to this study given its known role in T and B cell survival and specific published results vis-à-vis Gimap5 (Chen et al. *J Exp Med*, 2011). The authors' own data renders them a bit equivocal over engagement with this topic (see lines 88-92). At present, this reader feels he is being led in and out of a cul-de-sac on this subject. Some sharper writing/surgery is needed here (and in relation to what may be best addressed about this in the Discussion) in order to hold to the main message of the paper.

4. The molecular proposal for Gimap5 associated with vesicles such as MVBs sequestering and restraining GSK3 β is an interesting idea with origins within the GSK3 field (as reviewed in quoted ref 24). The data presented, however, only begin to address the idea. Issues such as whether the two molecules interact directly with each other and of their disposition with respect to the membranes of the MVBs, or other vesicles with which they are associated, need to be addressed. In connection with this, the authors draw attention to the potentially linked issue of the requirements for Ser389 phosphorylation in this system.

Other points

(list not exhaustive).

Line 21, Intro. As far as I know, human GIMAP5 polymorphisms have not been associated with risk for T1D. Indeed, ref. 3 explicitly rejects the hypothesis, as do Payn e et al. *Diabetes* (2004) 53:505. Only IA-2 autoantibodies have shown an association.

Line 25. References on autoimmunity/colitis. Missing reference: gastrointestinal autoimmune association with Gimap5 deficiency was first reported by Cousins et al. *Gastroenterology* (2006) 131:1475.

Line 53-56. Sentence should indicate that this refers to B lymphocytes.

Lines 75-6. Lifespan etc. Does this refer to previously published data? I see none here.

Line 77/Fig 1A,B. 'improved'. Relative to what? Cell numbers aren't necessarily the same as 'survival'. The cell numbers in the absence of Ova are still not as high as WT.

Fig 1A and legend. P values reported with many decimal places but not related specifically to the different comparisons.

Fig. 1C. Visual presentation of data marred by different numbers of events in the different panels. Tabulated data would be as much use as this.

Line 103/Fig S2C. No data on total S6K are presented.

Line 104-5/Fig 2A. Quality of Fig 2A is poor. Are the two panels in fact different areas of the same gel or are they not (as presented it looks as if they are not)? If not, then the data are of dubious validity unless very strict adherence to exposure times etc. has been applied. The actin controls do not add to confidence.

Fig 2A/C. Noticeable inconsistency in detection of c-Myc in unstimulated WT cells.

Fig 2C data panel. How have the data been normalised? Is this pairwise relative to five different WT controls? Fairly pointless as a panel rather than just stating the relative increase in sph/sph mice (with stats).

Fig 2G. Are these data from one experiment as a representative or do the stats encompass the 3 experiments? The 'Similarity dilate' parameter could do with some explanation.

Line 106-7/Fig 2B. myc transcription. mRNA levels are not a direct measure of transcription – there are other factors influencing this parameter.

Line 127/Fig 2C. 'restored'? Restored to what? The WT level is not reached. 'Increased' would be more accurate.

Line 130. I think this should say Fig.S1F.

Line 155-156. Please distribute ref. numbers between the points made rather than grouping them. In fact, ref 25 as given appears to be incorrect. I think that the paper that should have been quoted is Sutherland et al. (1993) Biochem J. 296:15.

Line 157. It would be helpful to re-quote ref.17 after MVBs.

Line 160-1, Fig 3A. The data as displayed/presented don't convince me that I am looking at 'high colocalisation'. Since the sph/sph mice are Gimap5-negative then the data should reflect zero vesicles positive for this. The 'Bright detail parameter is in need of explanation.

Fig 3B. Where is the statistical treatment of these data? Is the GSK3 β MFI a measure of individual vesicle intensity?

Lines 167-171. Fig. 3C. I am not sure of the data-value of this part. I see one yellow (co-localising) punctum against ca. 8 non-colocalised. It is not clear what the lower sph/sph pictures are meant to convey.

Line 182-3. There is a hint that S389 phosphorylation is increased at day 1. Is this a consistent observation?

Lines 184-185. p38 MAPK. Data not shown. Should this be presented?

Fig 3F. Was there no variance in WT data?

Line 189-191/Fig 3H. "DNA damage.....was reduced to control levels.....". It was reduced, but whether that is to control levels is doubtful.

Line 193. To what does 'these data' refer?

Lines 201-2/Fig 4A,B. Percentage of T cells within the spleen is a problematic measure to consider in this context. The proliferative phenotype associated with lymphopenia-associated colitis is accompanied by substantial changes in the spleen including large increases in myeloid cell populations. These inevitably distort the percentages of lymphocytes relative to WT controls. Numbers of lymphocytes in spleen is a more informative measure.

Fig 5A. As above, numbers of cells in spleen would be a more meaningful measure.

Fig 6C/D: catenin blots not convincing; in D, actin controls suggest that Gimap5^{-/-} LiCl⁺ track contains more protein than others.

Fig 7A-D. More detail is needed of the design of this experiment, and the collection and analysis of the data. I was taught that it is not possible to obtain standard deviations from a sample of n=2. Is that really what is meant? Surely substantial numbers of cells were analysed per point. Perhaps the statement at the end of the legend is meant to apply to the entire figure and not just 7J. This legend needs clarification throughout.

Fig S5C. These images, I suppose, are controls associated with Fig 4F,G. It seems absurd to separate these from their partners and segregate them into the supplementary bin.

Fig S6A. I don't understand this figure. I believe I can see 6 curves while the key only identifies 5. The colour-coding isn't good either, such that I can't figure out which curve is which. Are all groups tamoxifen-treated? Couldn't it be important that the sph/sph, GSK3wt cells are showing the highest levels of GSK3 β ?

Fig S6B. A plot of number would be better than percentage; as in C.

Fig S7. We have the appearance of the Gimap5^{+/-} genotype.

Line 237 and following. The human data is interesting but it should be pointed out that the pedigree data does not yet seem sufficient to associate the phenotype definitively with Gimap5.

Line 387-388. 'All mice used were generated on a C57BL/6 background and confirmed by whole genome SNP analysis'. Is this true? Or do you mean 'All strains/lines of mice'?

Line 400 et al. Check that origins of centrally important antibody reagents (e.g. for human and mouse Gimap5, for GSK3 etc.) are given clearly, with refs. If appropriate.

Real time PCR. Is the level of L32 RNA in mouse lymphocytes unaffected by T cell activation?

Reviewer #3 (Remarks to the Author):

Patterson and co-workers focus on mice deficient for Gimap5 which have reduced CD4T cells ad B

cells due to combined effects on survival and proliferation. Gimap 5 is needed for TCR induced inactivation of GSK3b which has b-catenin and NFAT signalling as downstream effectors. Interestingly pharmacological inhibition of GSK3 in Gimap5 deficient cells can restore proliferation, restore Treg numbers and overcome pathology in Gimap5 deficient mice. Authors report on one patient case with GIMAP5 deficiency where GSK3 inhibition also partially restored proliferation defects

In general the work is well done, but it remains unclear how GSK3 mediates its effects. GSK3b has many targets, including metabolic targets, C-Myc, Wnt-b-catenin and NFATc. A few simple Wnt and NFAT reporter gene experiments can address these important mechanistic experiments that are now lacking. In addition, it is unclear if the effects observed are exclusively mediated by GSK3b or also by GSK3a. The title also is ambiguous in this respect.

Other remarks:

In Fig 2, is PMA/Ionomycin treatment capable of bypassing the proliferation and c-Myc induction defects?

In Fig 3C the individual Gimap5 and GSK3b spots do not always co-localize, what percentage does (quantification)?

Comparing GSK3b kinase activity, how much is kinase activity inhibited after TCR/CD3/CD28 ligation vs. Pharmacological inhibition, which usually (at least with LiCl) is not complete

In Fig 4, the improvement of pathology (preventing colitis) largely depends on the number of Tregs. Therefore Treg numbers need to be included in panel 4A

The human patient does not have a marked CD4 T cell defect (273) but CD8 cells (30) are much more reduced. This is different from the mouse model. Explanation? Are CD8 T cell responses and proliferation also rescued by targeting GSK3b?

The discussion on GSK3 and Wnt signalling in TH17 and Tregs is incomplete. Several papers published in top journals showing that Wnt signalling activates Th17 responses and inhibits Treg responses are not cited nor discussed.

Specific responses to the reviewers' comments:

We thank the reviewers for their overall positive and constructive reviews. We believe that their insight and comments have helped to further improve the overall quality of our manuscript. Please below find our responses to their specific concerns. All textual changes in the revised manuscript are marked in red. We have additionally added a model figure (Fig. 9) to summarize our findings regarding the effects of Gimap5-deficiency in relation to GSK3 β regulation in the discussion (page 24).

Rebuttal Fig. 1 (manuscript Fig.9): *Gimap5* controls regulation of GSK3 β in T cells through vesicular sequestration and affects both the 1) (early) transcriptional program required for T cell growth, and 2) the late stage nuclear translocation of P-Ser³⁸⁹ GSK3 β required for the DNA damage response during cycling. *Gimap5*-deficient CD4⁺ T cells, fail to inhibit GSK3 β leading to a failed transcriptional program and increased DNA damage during T cells proliferation.

Reviewer #1:

1. They convincingly demonstrate that GSK3 β and *Gimap5* colocalize but it remains unclear how this facilitates phosphorylation. Do the authors think that *Gimap5* functions as a scaffolding molecule and facilitates GSK3 β phosphorylation through p38. At least some discussion would be helpful.

Although outside of the scope of the current manuscript, we agree with reviewer that this is an important question that remains to be answered. Interestingly, our new studies further refined the localization of

Rebuttal Fig. 2 (manuscript Fig. 3E-I): (E) Localization of total and P-Ser³⁸⁹ GSK3 β in WT or *Gimap5*^{spH/spH} CD4⁺ T cells after 2 days stimulation with α CD3/ α CD28 (n=3). (F) Nuclear localization and (H) expression of total GSK3 β . (G) Nuclear localization and (I) expression of P-Ser³⁸⁹ GSK3 β . ImageStream data represents average values of >500 CD4⁺ T cells per experiment; all experiments were performed at least three times. Statistical significance is determined by Student's two-tailed test.

phosphorylated GSK3 β and show that the phosphorylation of GSK3 β at Ser³⁸⁹ is predominantly present in the nucleus in line with previous observations. Importantly, overall GSK3 β nuclear translocation as well as P-Ser³⁸⁹ GSK3 β is markedly reduced in Gimap5-deficient mice. These observations point to a critical role for Gimap5 in the sequestration/trafficking of GSK3 β ultimately facilitating phosphorylation and translocation of GSK3 β in the nucleus. Importantly, in the absence of Gimap5, reduced P-Ser³⁸⁹ GSK3 β can still be observed in vesicle-like structures that are predominantly outside of the nucleus (Fig.4E), suggesting the GSK3 β Ser³⁸⁹ phosphorylation at day 2 may occur in unique intracellular compartments. Thus, we propose a model in which Gimap5 facilitates GSK3 β Ser389 phosphorylation and nuclear localization that is required for the late stage DNA damage response required for CD4⁺ T cell undergoing cycling. The exact molecular underpinnings of these observations, however remain to be defined in detail. We have added the data in the manuscript (new figure 3E-I) and expanded on this model in the discussion.

2. More background information on how unopposed GSK3 β activity induces DNA damage and T cell loss would be helpful.

While different studies indicate that pharmacological inhibition of GSK3 or shRNA-specific knockdown of GSK3 β promotes DNA damage repair (Yang *et al.*, *Neuro-Oncology*, 2011), the exact molecular mechanisms by which GSK3 β regulates the DNA damage response unfortunately remain mostly elusive to date. The studies by Thornton *et al.* using a Ser³⁸⁹Ala knockin mouse model suggest that the P-Ser³⁸⁹ of GSK3 β , is important for lymphocyte fitness and survival. In addition, evidence suggest that the GSK3-mediated regulation of the transcriptional program (e.g. c-Myc, NFATc1) contributes to the optimal DNA repair response. For instance, consistent with our observations, inhibition of NFAT nuclear translocation impairs DNA repair upon UV irradiation (Canning *et al.*, *J Mol Histol.*, 2006; Yarosh *et al.*, *J Invest Dermatol.*, 2005). We included these references in the text (line 363-370).

3. The authors do not describe b-catenin data of Figure 6d in the result section. Also it looks like b-catenin levels are not really different, contrary to the interpretation of the authors in discussion section (line 289-290).

We agree with the reviewer that the overall levels of β -catenin, as assessed by immunoblot analysis, are low and therefore any change in β -catenin levels are difficult to interpret. In our revised manuscript, we have excluded the human β catenin data and softened the overall conclusion on β -catenin while the relevant mouse data has been moved to Supplemental Fig.4A-C.

4. In line 130, fig.S1G is incorrect. It should be fig.S1F.

We have corrected this mistake in the revised version.

Reviewer #2 (Remarks to the Author):

Major points

1. The argument and data presented surround the effects of Gimap5 deficiency on events following conventional T cell activation via the TCR. What is not clear is how this relates to the primary aspect of the phenotype associated with Gimap5 deficiency, namely the in vivo T cell lymphopenia. Gimap5-deficient

animals fail to maintain a population of naïve, resting T cells. Part of this phenotype probably corresponds to a reduced thymic output of T cells and another part to a failure to maintain T cells in the resting condition (this point has been explicitly demonstrated in experiments on *Gimap1*-related lymphopenia). Neither of these stages corresponds obviously to explicit antigen-driven T cell responses. Late SP thymocytes and resting T cells express notably high levels of *Gimap5*. The authors should address this issue.

Our data depicted in manuscript Fig.1 indicates that CD4⁺ T cells levels in *Gimap5*-deficient mice are largely maintained in the absence of cognate antigen (*Gimap5*^{sp^h/sp^h; *rag2*^{-/-}; OT-II mice given water without ovalbumin); suggesting that activation-induced cell death (particularly post weaning in which the animals adapt to changing/new microbial flora) serves as an important contributor of the CD4⁺ T cell-specific lymphopenia. But the reviewer is correct in that the CD4⁺ T cell levels do not entirely reach WT levels (Manuscript Fig.1a) and thus, thymic output or reduced survival of peripheral resting CD4⁺ T cells as suggested by the reviewer could present a contributing factor of lymphopenia. We have now included studies to directly assess whether thymic output/survival or peripheral survival of CD4⁺ T cells are affected}

Rebuttal Fig.3 (Manuscript Fig.S1A-C and Fig1.D): No defect in thymic survival or output. (A) Number of single positive CD4⁺/CD8⁻ thymocytes surviving *ex vivo* when cultured with IL-7 (5 ng/mL) (n=6). (B) Percentage of splenic CD4⁺ T cells in WT and *Gimap5*^{sp^h/sp^h mice, and (C) frequency of recent thymic emigrants (RTEs) (CD24^{hi}) and naïve (CD62L^{hi}CD44^{lo}) cells among peripheral CD4⁺ T cells from three-week-old mice (n=6). Data depict mean +/- SEM. (D) Calcium flux in CD4 SP thymocytes stimulated with 1 µg/mL αCD3/αCD28. Representative of 3 independent samples (E) Number of peripheral CD4⁺ T cells surviving *ex vivo* when cultured with IL-7 (5 ng/mL) (n=3).}

in *Gimap5*-deficient mice. Importantly, we observed a normal survival of thymic SP CD4⁺ T cells when cultured in the presence of IL-7 *ex vivo* (new Fig.S1A), while quantification the number of recent thymic emigrants (CD4⁺, CD24^{hi}) in the spleen of 3-week old WT and *Gimap5*^{sp^h/sp^h mice, revealed no differences in thymic output between WT and *Gimap5*^{sp^h/sp^h mice (new Fig.S1B, C). This is also in line with the GIMAP5 patient, who exhibits a normal proportion of CD31⁺ T cells that are indicative of recent thymic emigrants (see supplemental patient description). In contrast, peripheral CD4⁺ T cells showed a slight reduction in survival when cultured in the presence of IL-7 *ex vivo* (new Fig.1D). Together these studies suggest that in line with our previous studies (Barnes *et al.*, *Jl*, 2009), thymic development and survival of CD4⁺ T cells is largely unaffected. In contrast however, our studies do indicate a slight impediment in the survival of peripheral CD4⁺ T cells in the presence of IL-7. These studies are described on page 6 and 7 of our revised manuscript.}}

2. The authors have conducted the majority of their studies using T lymphocytes from 3-wk old mice, when the mutant mice possess a phenotypically fairly normal T cell population, in contrast to later time points. The assumption that the T lymphocytes at this stage are 'normal', however, is not without risk. For instance, *Gimap5*-deficient SP thymocytes have a survival deficiency and this may extend into the cell stage they are studying. In the same vein, calcium responses of *Gimap5*-deficient SP thymocytes are reported to be down relative to WT (Ilangumaran et al. Mol Immunol 2009). A system in which *Gimap5* could be electively deleted in lymphocytes would have been preferable.

See comments above (1): Our new studies indicate that *Gimap5*^{sph/sph} mice exhibit no apparent deficiency in thymic survival while displaying a normal thymic output of SP CD4⁺ thymocytes (see rebuttal Fig. 3). In addition to these studies, we tested calcium responses in SP CD4⁺ thymocytes upon activation with CD3/CD28 or ionomycin. Consistent with the normal survival and thymic output of SP CD4⁺ thymocytes, we observed no differences between the release of intracellular calcium between WT and *Gimap5*^{sph/sph} SP CD4⁺ thymocytes (rebuttal Fig. 3D; not included in the manuscript). These findings deviate from the observations in the *lyp/lyp* rat model. Thus, while the use of conditional *Gimap5* KO would be ideal, we believe that the 3-week old *Gimap5*^{sph/sph} mice offer a reliable model to study peripheral CD4⁺ T cells at a relative naïve stage.

3. *Mcl-1*. As a target of *GSK3β*, *Mcl-1* is of interest to this study given its known role in T and B cell survival and specific published results vis-à-vis *Gimap5* (Chen et al. J Exp Med, 2011). The authors' own data renders them a bit equivocal over engagement with this topic (see lines 88-92). At present, this reader feels he is being led in and out of a cul-de-sac on this subject. Some sharper writing/surgery is needed here (and in relation to what may be best addressed about this in the Discussion) in order to hold to the main message of the paper.

We have improved the description of the *Mcl-1* data in the results section (page 8 and 9) and further discussed the potential impact of reduced *Mcl-1* levels in the discussion. While our findings (i.e. reduced *Mcl-1* levels in CD4⁺ T cells) are consistent with the study of Chen et al. (J Exp Med, 2011), deletion of the pro-apoptotic molecules *Bax/Bak*, normally balanced by *Mcl1* expression, did not improve overall survival of CD4⁺ T cells. Thus, these findings suggest a limited impact of the reduced *Mcl-1* levels on the overall peripheral survival of CD4⁺ T cells.

4. The molecular proposal for *Gimap5* associated with vesicles such as MVBs sequestering and restraining *GSK3β* is an interesting idea with origins within the *GSK3* field (as reviewed in quoted ref 24). The data presented, however, only begin to address the idea. Issues such as whether the two molecules interact directly with each other and of their disposition with respect to the membranes of the MVBs, or other vesicles with which they are associated, need to be addressed. In connection with this, the authors draw attention to the potentially linked issue of the requirements for Ser389 phosphorylation in this system.

The vesicle association of GSK3 β (colocalized with Gimap5) is indeed a surprising and novel observation and as a mechanism of GSK3 β -regulation. We have now further defined the GSK3 β^+ /Gimap5 $^+$ double-positive vesicles in CD4 $^+$ T cells using ImageStream analysis. Interestingly, our studies revealed limited expression of Rab5, Rab7 and Lamp1 $^+$ in Gimap5 $^+$ /GSK3 β^+ vesicles. As far as a potential mechanism, our initial experiments performing co-immunoprecipitation studies, yielded no evidence for a direct interaction between Gimap5 and GSK3 β through studies in resting or TCR-activated CD4 $^+$ T cells (results not shown). The timing and technical limitations associated with these Co-IP experiments, however, does not (yet) exclude this possibility. Importantly, further definition by ImageStream analysis reveals the P-Ser³⁸⁹ GSK3 β to be almost exclusively present in the nucleus of WT CD4 $^+$ T cells, while *Gimap5*^{sp^h/sp^h CD4 $^+$ T cells exhibit markedly reduced P-Ser³⁸⁹ GSK3 β levels (see new manuscript Fig.3E-I and comments above for reviewer 1). These findings suggest that Gimap5 facilitates the P-Ser³⁸⁹ and distribution of GSK3 β in the nucleus that is required for the DNA damage response occurring during T cell cycling. The exact molecular mechanisms underlying these observations are beyond the scope of the current manuscript and will be a key focus in future studies in our laboratory.}

Rebuttal Fig.4 (manuscript Fig.S5): Vesicular localization of Gimap5 and GSK3 β . (A) Representative images of CD4 $^+$ T cells derived from WT mice either resting or stimulated 24h with α CD3/ α CD28. (B) Number of Gimap5 $^+$ and Gimap5 $^+$ Lamp1 $^+$ vesicles. (C) Colocalization of GSK3 β with Gimap5 or Lamp1 in activated CD4 $^+$ T cells. (D,E) Expression of Rab5, Rab7, and Lamp1 in Gimap5 and GSK3 β DP vesicles as defined by (D) spot area and (E) Intensity of Rab5, Rab7, or Lamp1 in DP vesicles. Bars depict mean \pm SD (n=4). Each ImageStream data point represents average values of >500 CD4 $^+$ T cells.

Other points

Line 21, Intro. As far as I know, human GIMAP5 polymorphisms have not been associated with risk for T1D. Indeed, ref. 3 explicitly rejects the hypothesis, as do Payne et al. Diabetes (2004) 53:505. Only IA-2 autoantibodies have shown an association.

The reviewer is correct that the relevant SNP (rs6598) studied by Shin et al. shows significant association with tyrosine phosphatase-related islet antigen 2 (IA-2) auto-antibodies. Thus, as suggested by Shin et al.,

we rephrased this into “..associated with islet autoimmunity in type 1 diabetes” (page 3, line 21, page 28, line 445). Notably, this specific SNP was underpowered and assessed in only 32 type 1 diabetic subjects in the study by Payne *et al.*, making it difficult to directly compare both studies.

Line 25. References on autoimmunity/colitis. Missing reference: gastrointestinal autoimmune association with Gimap5 deficiency was first reported by Cousins et al. Gastroenterology (2006) 131:1475.

We apologize for this oversight; this citation has been added (page 3, line 25).

Line 53-56. Sentence should indicate that this refers to B lymphocytes.

We have modified this sentence to: “GSK3 β by phosphorylation of Ser³⁸⁹ is essential for lymphocyte viability upon double-stranded DNA breaks (DSBs) observed during Variable, Diversity, and Joining (V(D)J) recombination during thymic T cell development or in B cells undergoing immunoglobulin class switch recombination (CSR).” Line 53-56.

Lines 75-6. Lifespan etc. Does this refer to previously published data? I see none here.

This is based on our personal experiences keeping *Gimap5^{sph/sph}; Rag2^{-/-}* OT-II breeders. Mice were kept generally up to 6 months and showed similar survival compared to *Gimap5^{WT} Rag2^{-/-}*, OT-II mice. We observed no spontaneous colitis in these mice as assessed by histology. Since the reference of the lifespan observation is anecdotal, and not compiled in a data plot, we removed this statement from the text.

Line 77/Fig 1A,B. ‘improved’. Relative to what? Cell numbers aren’t necessarily the same as ‘survival’. The cell numbers in the absence of Ova are still not as high as WT.

The text is now modified to “Moreover, their T cell compartment is largely maintained in the absence of antigen and CD4⁺ T cells remain predominantly naïve (CD44^{lo};CD62^{hi}) (Fig.1A,B). On the other hand, compared to the *Gimap5*-sufficient controls, *Gimap5^{sph/sph};Rag2^{-/-}*;OT-II mice exposed to ovalbumin in drinking water, exhibited a reduced CD4⁺ T cell population with an increased proportion of remaining CD4⁺ T cells displaying a memory-like phenotype (CD44^{hi};CD62^{Lo}) (Fig.1A,B)” (line 93-99). The issue with peripheral survival is now also addressed with added studies assessing ex vivo survival of peripheral CD4 T cells with IL-7 (see above, reviewer 2, comment 2).

Fig 1A and legend. P values reported with many decimal places but not related specifically to the different comparisons.

P values are now directly indicated in the figures.

Fig. 1C. Visual presentation of data marred by different numbers of events in the different panels. Tabulated data would be as much use as this.

Figure 1c is replaced by a bar graph in the revised version.

Line 103/Fig S2C. No data on total S6K are presented.

Total S6K is now included in Fig.S3D.

Line 104-5/ Fig 2A. Quality of Fig 2A is poor. Are the two panels in fact different areas of the same gel or are they not (as presented it looks as if they are not)? If not, then the data are of dubious validity unless very strict adherence to exposure times etc. has been applied. The actin controls do not add to confidence.

The immunoblot indeed represents different areas of the same gel. Nonetheless, to avoid any ambiguity, a new immunoblot showing all samples in a single gel is now included in Figure 2A.

Fig 2A/C. Noticeable inconsistency in detection of c-Myc in unstimulated WT cells.

See above; Fig.2a has been replaced with a new immunoblot and shows consistent c-Myc data compared to 2C. In the previous Immunoblot, the induction of c-Myc was less robust (also in WT) and therefore the blots were longer exposed, explaining the differences in c-Myc at baseline (unstimulated conditions) between 2A and 2C.

Fig 2C data panel. How have the data been normalised? Is this pairwise relative to five different WT controls? Fairly pointless as a panel rather than just stating the relative increase in sph/sph mice (with stats).

We thank the reviewer on suggesting an alternative method to present the data, which are pairwise. Data points were generated by measuring the fold increase in the proportion of c-Myc phosphorylation in *Gimap5^{sph/sph}* CD4⁺ T cells compared to WT CD4⁺ T cells. The five data points correspond to five independent experiments; each comparison was performed on samples of a single blot to avoid introducing error from transfer efficiency, labeling efficiency, exposure, and stripping efficiency. Notably, in every experiment/blot we ran, the proportion of phosphorylated c-Myc in *Gimap5^{sph/sph}* CD4⁺ T cells was significantly higher than in WT CD4⁺ T cells. This is indicative of increased GSK3-mediated phosphorylation.

Fig 2G. Are these data from one experiment as a representative or do the stats encompass the 3 experiments? The 'Similarity dilate' parameter could do with some explanation.

The data is representative of 3 independent experiments included, (*n* of 2 per experiment, for a total of *n*=6). This is now further clarified in the legend of Fig.2. The similarity dilate represents the log transformed Pearson's Correlation Coefficient and is a measure of the degree to which two images are linearly correlated within a masked region. In this case both NFATc1 and DAPI were masked individually in live CD4⁺ T cells (>500 cells per sample for all experiments). We added the explanation to the material and methods (Page 33, Line 529-533).

Line 106-7/ Fig 2B. myc transcription. mRNA levels are not a direct measure of transcription – there are other factors influencing this parameter.

While mRNA stability and other factors may also affect the total mRNA level, this data shows that the failure to accumulate cMyc protein was not due to a failure to transcribe the *myc* gene. We have modified the language.

Line 127/Fig 2C. 'restored'? Restored to what? The WT level is not reached. 'Increased' would be more accurate. Line 130. I think this should say Fig.S1F.

Thank you; this has been corrected.

Line 155-156. Please distribute ref. numbers between the points made rather than grouping them. In fact, ref 25 as given appears to be incorrect. I think that the paper that should have been quoted is Sutherland et al. (1993) Biochem J. 296:15.

Thank you; the suggested changes have been included.

Line 157. It would be helpful to re-quote ref.17 after MVBs.

The suggested reference is now also included here.

Line 160-1, Fig 3A. The data as displayed/presented don't convince me that I am looking at 'high colocalisation'. Since the sph/sph mice are Gimap5-negative then the data should reflect zero vesicles positive for this.

The reviewer is correct and this is indeed an overstatement. In fact it is important to note that not all GSK3 β ⁺ vesicles are Gimap5 positive. We believe that prior to localization to Gimap5, GSK3 β likely must progress through a stepwise process through several different vesicular compartments (early endosomes, etc.). The vesicular distribution of GSK3 β varies between cells and is dependent upon activation state. We have clarified this statement and in new Fig.S5, we further define of Gimap5⁺/GSK3 β ⁺ DP vesicles and their coexpression with endosomal/lysosomal vesicle markers. We also now quantified the % of Gimap5⁺ vesicles in human and mouse CD4⁺ T cells that are positive for GSK3 β in the text (Line 197 and Line 322).

The 'Bright detail parameter is in need of explanation.

The Bright Detail Similarity is a standard statistical analysis performed by IDEAS software that represents the log transformed Pearson's correlation coefficient of the localized bright spots with a radius of 3 pixels or less within the masked area in the two input images. Our studies in this particular experiment were masked on Gimap5 spots and GSK3 β spots. In general values <1 suggest no colocalization. This is now described in the material and methods on page 33, line 540-543.

Fig 3B. Where is the statistical treatment of these data? Is the GSK3 β MFI a measure of individual vesicle intensity?

P values have been added and indeed the GSK3 β MFI is the average fluorescent intensity for individual spots.

Lines 167-171. Fig. 3C. I am not sure of the data-value of this part. I see one yellow (co-localising) punctum against ca. 8 non-colocalised. It is not clear what the lower sph/sph pictures are meant to convey.

The figure is meant to convey a higher resolution Image, while the Image stream provides the quantitative analysis of Gimap5/GSK3 β + spots. The lower panel represent a control for the specificity of the Gimap5 antibody as one would expect to include in this type of experiment.

Line 182-3. There is a hint that S389 phosphorylation is increased at day 1. Is this a consistent observation?

GSK3 β phosphorylation at S389 at day 1 appears similar between WT and *Gimap5^{sp^h/sp^h}* CD4⁺ T cells in the figure shown (and other blots) and appears unchanged to *Gimap5^{sp^h/sp^h}* levels in resting conditions or at day 2.

Lines 184-185. p38 MAPK. Data not shown. Should this be presented?

p38 expression and phosphorylation is shown in Fig. 3D, blots 4-5 as well as Fig.S3B and Fig.S7C. We have added the Figure notations to clarify this.

Fig 3F. Was there no variance in WT data?

While there was variance in the WT data, error bars would be shorter than the height of the symbol. In these cases, Prism simply does not draw the error bars.

Line 189-191/Fig 3H. "DNA damage.....was reduced to control levels.....". It was reduced, but whether that is to control levels is doubtful.

This has been modified and now simply states "reduced".

Line 193. To what does 'these data' refer?

The data shared in this section, as an aggregate.

Lines 201-2/Fig 4A,B. Percentage of T cells within the spleen is a problematic measure to consider in this context. The proliferative phenotype associated with lymphopenia-associated colitis is accompanied by substantial changes in the spleen including large increases in myeloid cell populations. These inevitably distort the percentages of lymphocytes relative to WT controls. Numbers of lymphocytes in spleen is a more informative measure.

Gimap5^{sp^h/sp^h} mice are consistently smaller than WT littermates, even pre-weaning (Barnes, *Jl*, 2010). This corresponds to an overall numeric reduction in cellularity within lymphoid organs. Furthermore, the LiCl treatment affects weight of mice (WT and *Gimap5^{sp^h/sp^h}*) — something that cannot be accounted for in the untreated controls. We therefore believe that showing frequencies of CD4⁺ T cells and B cells for this experiment is a more accurate measurement.

Fig 5A. As above, numbers of cells in spleen would be a more meaningful measure.

Fig 6C/D: catenin blots not convincing; in D, actin controls suggest that Gimap5^{-/-} LiCl⁺ track contains more protein than others.

Given the low β -catenin levels detected in T cells and the associated difficulty to draw conclusive data from these WBs, we moved the β -catenin immunoblot to the supplemental data and are careful in our interpretation of these data.

Fig 7A-D. More detail is needed of the design of this experiment, and the collection and analysis of the data. I was taught that it is not possible to obtain standard deviations from a sample of $n=2$. Is that really what is meant? Surely substantial numbers of cells were analyzed per point. Perhaps the statement at the end of the legend is meant to apply to the entire figure and not just 7J. This legend needs clarification throughout.

The individual data points present the mean values of >500 cells per sample (now clarified in the legend). Moreover, the data are now depicted as individual data points representing the average for each sample.

Fig S5C. These images, I suppose, are controls associated with Fig 4F,G. It seems absurd to separate these from their partners and segregate them into the supplementary bin.

The idea was to maximize histology panels so that the details in tissues actually showing pathology remain clear. Nonetheless, we have added these back in (current) Figure 5 in the text.

Fig S6A. I don't understand this figure. I believe I can see 6 curves while the key only identifies 5. The colour-coding isn't good either, such that I can't figure out which curve is which. Are all groups tamoxifen-treated? Couldn't it be important that the sph/sph, GSK3wt cells are showing the highest levels of GSK3 β ?

The sixth curve was a second *Gimap5^{sph/sph}Gsk3b^{fl/fl}* mouse. It has been removed. When comparing *Gsk3b^{wt/wt}* cells, the increased GSK3 β MFI in *Gimap5^{sph/sph}* CD4⁺ T cells is reflective of progressive shift of *Gimap5*-deficient CD4⁺ T cells to a memory like phenotype (CD44^{hi}CD62L^{lo}). Memory cells in general have higher expression of GSK3 β independent of *Gimap5* expression; GSK3 β levels in WT and *Gimap5*-deficient CD4⁺ T cells are equivalent in CD44^{hi}CD62L^{lo} CD4⁺ T cells. All groups are tamoxifen treated.

Fig S6B. A plot of number would be better than percentage; as in C.

We have included numbers instead of frequencies in the B cell graph (Fig.S9B).

*Fig S7. We have the appearance of the *Gimap5*^{+/-} genotype.*

Correct, the *GIMAP5^{+/-}* genotype (parent) was also used in Fig.7 and Fig.8H-J. (Current) Line 308 also clearly states that “we compared T cell proliferation from the patient with his heterozygous mother.”

*Line 237 and following. The human data is interesting but it should be pointed out that the pedigree data does not yet seem sufficient to associate the phenotype definitively with *Gimap5*.*

Given 1) the complete loss of GIMAP5 protein, 2) the observed similarity in molecular defects with *Gimap5*-deficient mice, 3) the absence of other candidate variants that could explain the deficiency, and 4) the use of GIMAP5 heterozygous parents that serve as ideal genetic controls, we believe we don't overreach with regard to our statements. More importantly, we believe we have been very careful in our overall conclusions in the manuscript by stating “we show that a human patient carrying a GIMAP5 loss-

of-function mutation exhibits similar deficiencies, including lymphopenia, recurrent thrombocytopenia, and impaired T cell function. T cells of this patient exhibit reduced T cell expansion and c-Myc expression that can be corrected *in vitro* with GSK3 inhibitors.”

Line 387-388. ‘All mice used were generated on a C57BL/6 background and confirmed by whole genome SNP analysis’. Is this true? Or do you mean ‘All strains/lines of mice’?

This has been modified and instead of mice, now states “all mouse strains”.

Line 400 et al. Check that origins of centrally important antibody reagents (e.g. for human and mouse Gimap5, for GSK3 etc.) are given clearly, with refs. If appropriate. Real time PCR. Is the level of L32 RNA in mouse lymphocytes unaffected by T cell activation?

Antibody information has been provided in the Materials and Methods section. Yes, L32 (or RPL32) RNA is unaffected by T cell activation, and is frequently used as a housekeeping gene (Reis *et al.*, *Nat Immunol.*, 2013; Wang *et al.*, *Nat Immunol.*, 2014, Andris *et al.*, *Front Immunol.*, 2017).

Reviewer #3 (Remarks to the Author):

In general, the work is well done, but it remains unclear how GSK3 mediates its effects. GSK3b has many targets, including metabolic targets, C-Myc, Wnt-b-catenin and NFATc. A few simple Wnt and NFAT reporter gene experiments can address these important mechanistic experiments that are now lacking.

We thank the reviewer for his constructive feedback. The reporter gene experiments would indeed be an elegant way to complement our findings with regard to the GSK3 targets c-Myc, NFATc1 and/or β -catenin. However, based on our experience using lentiviral vectors as well as transfection approaches in Gimap5-deficient CD4⁺ T cells, we observed that the inherent fragility of these cells (likely due to their sensitivity to DNA damage) has made it impossible to reliably introduce exogenous vectors and, for instance, perform gain of function studies for specific transcription factors. Nonetheless, blocking NFATc1 dephosphorylation (calcineurin inhibitor) in T cells causes a major loss of T cell proliferation (manuscript Fig.S4D), confirming the critical role of NFAT in T cell proliferation. We have also performed studies to address whether GSK3 β vesicular sequestration involved Wnt signaling. Specifically, we incubated WT CD4⁺ T cells directly with Wnt3a and/or tested the effect of blocking Wnt secretion by incubating CD4⁺ T cells with α CD3/CD28 in the presence/absence of IWP-2 (a Wnt secretion inhibitor). Subsequently, we assessed GSK3 β sequestration by Imagestream and determined the ability of WT CD4⁺ T cells to proliferate. Neither Wnt3a nor IWP-2 affected the vesicular sequestration in WT CD4⁺ T cells or impacted their proliferation (Fig.S6A-D). Moreover, no significant differences in *Wnt3a* expression was observed between WT and *Gimap5*^{sph/sph} CD4 T cells at resting or activated conditions. These results are included in (Fig.S6) and discussed on page 15, line 208-215.

Rebuttal Fig.5 (manuscript Fig.S6): No observed effect of Wnt signaling on vesicular localization of GSK3β or proliferation. (A-C) Vesicular localization of GSK3β in WT CD4⁺ T cells stimulated 24h with 200 ng/mL Wnt3a or αCD3/αCD28 +/- 2 μM IWP-2 as measured by (A) number of GSK3β⁺ spots, (B) GSK3β vesicular intensity, and (C) size of GSK3β⁺ spots (n=3). Each ImageStream data point represents average values of >500 CD4⁺ T cells. (D) Proliferation of WT CD4⁺ T cells after 3d stimulation with αCD3/αCD28 +/- Wnt3a or IWP-2 (n=4). Data represent means +/- SD. (E) Wnt3a mRNA levels in resting and αCD3/αCD28-activated (24h) CD4⁺ T cells (n=9). Bars depict mean ± SEM. Statistical significance is determined by Student's two-tailed test.

In addition, it is unclear if the effects observed are exclusively mediated by GSK3β or also by GSK3α. The title also is ambiguous in this respect.

With regard to the specific role of GSK3α and GSK3β, it is predicted that a majority of their functions are redundant. However, our data shows that genetic deletion of GSK3β is sufficient to mitigate CD4⁺ T cell loss and development of colitis in *Gimap5^{sph/sph}* mice. While this does not rule out the possibility that GSK3α is also dysregulated in the absence of Gimap5, these data point to a key role of GSK3β in the underlying T cell-dependent pathology. This specificity could be related to the DNA damage response which in the current literature is predominantly associated with GSK3β.

Other remarks:

In Fig 2, is PMA/Ionomycin treatment capable of bypassing the proliferation and c-Myc induction defects?

PMA/Ionomycin results in similar molecular and cellular defects, including impaired proliferation, survival, and induction of c-Myc. Given the normal calcium flux in response to Ionomycin or αCD3/αCD28 (Fig.S4F) and the activation of ERK, JNK IκB, AKT and p38 (Fig.S3B) upon PMA/Ionomycin stimulation and AKT, p38, and GSK3β (S9) in Fig.S7A-D upon αCD3/αCD28 stimulation, it is likely that and Gimap5-associated defects occur further downstream.

Rebuttal Fig.6 (not included in the manuscript): WT and *Gimap5^{sph/sph}* CD4⁺ T cells stimulated with PMA (50 ng/mL) and Ionomycin (500 ng/mL) for 24h prior to lysis.

In Fig 3C the individual *Gimap5* and *GSK3b* spots do not always co-localize, what percentage does (quantification)?

The reviewer is correct and likely, prior to localization to *Gimap5*, *GSK3β* must progress through a stepwise process through several different vesicular compartments (early endosomes, etc.). The vesicular distribution of *GSK3β* varies between cells and is dependent upon activation state. In Fig. S5, we further define the *Gimap5*⁺*GSK3β*⁺ spots (Fig.S5) at 24h post stimulation and we also assess nuclear translocation of *GSK3β* which is primarily observed at a late stage (48h post stimulation) (Fig.3). In general we see a ~66% of *Gimap5*⁺ vesicles expressing *GSK3β* at 24 hours of activation for human CD4⁺ T cells, while in mice, the percentage of *Gimap5*⁺ vesicles expressing *GSK3β* is ~54%. These values have been included in the text (Line 197 and Line 322).

Comparing *GSK3b* kinase activity, how much is kinase activity inhibited after TCR/CD3/CD28 ligation vs. Pharmacological inhibition, which usually (at least with LiCl) is not complete

We show that *GSK3* inhibitors LiCl and BIO both significantly reduce phosphorylation of cMyc. Shown to in the figure below is the level of p-cMyc (T58) in α CD3/ α CD28 stimulated WT and *Gimap5*^{sp^h/sp^h} CD4⁺ T cells treated with either 2.5 mM LiCl or 100 nM BIO relative to untreated WT or *Gimap5*^{sp^h/sp^h} CD4⁺ T cells. Using cMyc phosphorylation at T58 as a readout of *GSK3* activity, these data show that *GSK3* activity is reduced by ~60% by 2.5 mM LiCl and 40% by 100 nM BIO (*n*=3 and *n*=10, respectively).

Rebuttal Fig.7 (not included in the manuscript): Level of p-cMyc (T58) in WT and *Gimap5*^{sp^h/sp^h} CD4⁺ T cells stimulated with α CD3 (1 μ g/mL) and α CD28 (2 μ g/mL) \pm 2.5 mM LiCl or 100 nM BIO relative to untreated WT and *Gimap5*^{sp^h/sp^h} CD4⁺ T cells (Data represents mean \pm SD; *n*=3 and *n*=10, respectively).

In Fig 4, the improvement of pathology (preventing colitis) largely depends on the number of Tregs. Therefore Treg numbers need to be included in panel 4A.

The frequency of Treg cells in the spleen of vehicle- and LiCl- treated *Gimap5*^{sp^h/sp^h} mice remains unaffected and is not different from WT mice in those conditions (new Fig.S8C). Instead, Fig. 5D demonstrates that the therapeutic effect predominantly works on their suppressive capacity. As published previously (Aksoylar *et al.*, *Jl*, 2012) and shown here, while *Gimap5*^{sp^h/sp^h} have similar frequencies of regulatory T cells, they have a markedly impaired suppressive capacity. We show that 4 week treatment

with LiCl restores *Gimap5*^{sph/sph} Treg function. This is consistent with previous reports of GSK3 inhibition potentiating Treg activity.

Rebuttal Fig.8 (manuscript Fig.S8C): (A) Frequency of regulatory T cells ($CD4^+CD25^{hi}Foxp3^+$) in the splenic $CD4^+$ T cell compartment of vehicle- and LiCl-treated WT and *Gimap5*^{sph/sph} mice ($n=11$; mean \pm SEM). Statistical significance is determined by ANOVA followed by Sidak's multiple comparisons test.

The human patient does not have a marked $CD4^+$ T cell defect (273) but $CD8^+$ cells (30) are much more reduced. This is different from the mouse model. Explanation? Are $CD8^+$ T cell responses and proliferation also rescued by targeting *GSK3b*?

Patient $CD4^+$ T cells are significantly reduced (246) compared to reference range for his age (610-1446). Expansion of both $CD4^+$ and $CD8^+$ T cells was rescued upon GSK3 inhibition with LiCl. This is actually remarkably similar to the *Gimap5*^{sph/sph} mouse. The mouse lacks peripheral $CD8^+$ T cells, while $CD4^+$ T cells are significantly reduced, but not absent. Given this phenotype, as well as the more severe reduction of $CD8^+$ T cells in the *GIMAP5* patient, it is reasonable to conclude that $CD8^+$ T cells are more susceptible to *Gimap5*-deficiency.

The discussion on GSK3 and Wnt signaling in *TH17* and Tregs is incomplete. Several papers published in top journals showing that Wnt signaling activates *Th17* responses and inhibits Treg responses are not cited nor discussed.

We have expanded the discussion and added additional references discussing the roles of Wnt signaling, GSK3, and β -catenin in $CD4^+$ T cell polarization, a topic that remains somewhat controversial. Loosdregt *et al.* report that addition of Wnt and inhibition of GSK3 *in vitro* inhibits Treg suppressive capacity (*Immunity*, 2013). Lee *et al.*, demonstrate that inhibition of Wnt signaling promotes *Th17* activity, while addition of exogenous Wnt or GSK3 inhibition reduce IL-17 production (*Euro J Immuno.*, 2012). Conversely, expression of constitutively active β -catenin promotes *Th17* polarization (Keerthivasan, *et al.*, *Sci Trans Med.*, 2014). Other studies suggest that inhibition of GSK3 promotes Treg activity (references 58,61,63). While the literature is ambiguous, our data support the idea that uncontrolled GSK3 β activity impairs Treg function while at the same time an increased *Th17* development is observed (Aksoylar *et al.*, *Jl* 2012). These additional papers are now referenced listed in the discussion (Line 422-424).

Reviewers' comments:

Reviewer #1 (Remarks to the Author):

The authors have appropriately addressed the issues that I have raised in my previous review.

Reviewer #2 (Remarks to the Author):

I welcome the additional experiments that have been performed and the improvements that have been made to this manuscript. The study by Patterson and colleagues now makes a good case for a strong regulatory influence of Gimap5 on the actions of GSK3b in T cells. It proposes that Gimap5 has effects both on the physical sequestration of GSK3b (in MVBs) – an inhibiting effect on GSK3b - and on the nuclear translocation of GSK3b – an effect that somehow may promote a productive T cell response (see lines 387-89). I don't think the precise linkage of these effects to S389 phosphorylation of GSK3b has been fully delineated at this stage.

In my opinion the manuscript could still do with some revision to correct inaccuracies, errors and improve presentation. Below I present my commentary which is in roughly line order. Items 1, 17, 18 raise points of clarification about the biology. I think the provision of a schematic diagram (Fig. 9) is helpful. This version, however, isn't entirely limp for me in its explanatory power. I have asked questions relevant to this (items 17-19).

1. Lines 66-82. The new data on ex vivo survival of thymocytes and (later) peripheral T cells are valuable additions. However, (and I am not wishing to belabour this point) there are in these lines some inaccuracies in overstating the degree of consistency with the rat *lyp/lyp* model. Line 68 – there is in fact evidence against progressive loss of peripheral CD4+ cells in the rat model (see Yale J-F, Grose & Marliss (1985) *Diabetes* 34:955: Time course of the lymphopenia in BB rats): lymphocyte numbers appear to start low and stay low. Similarly, the statement (line 80) that the new data on *sph/sph* RTEs 'are consistent with observations in the *lyp/lyp* rat' appears to overlook published data to the contrary (Zadeh et al. (1996) *Autoimmunity* 24:35; Ramanathan & Poussier (2001) *Immunological Reviews* 184:161). One has to conclude, in fact, that there are significant differences in detail between the Gimap5 *sph/sph* and the Gimap5 *lyp/lyp* models. Whether these differences are a matter of species difference or, alternatively, of the specific details of the genetic lesions under study, is so far unclear. There is a scatter of observations suggesting biological activity of fragments of Gimap5 (e.g. Sandal et al. (2003) *Mol. Biol. Cell* 14:3292). The three available mouse Gimap5 deficient strains and the rat model may differ in the persistence of cryptic levels of Gimap5 polypeptides with some activity.

2. Line 133. 'A marked reduction.....' Contrary to this statement, and after the updating of this figure, Fig 2 no longer shows a reduction of c-Myc in resting (unstimulated) *sph/sph* T cells. As one might expect, there is very little expression at all. Re-write required.

3. Line 150 – there is reference to resting T cells but Fig. S4A does not appear to show data for unstimulated cells, so this sentence should be modified. In the same sentence I think 'further indicating' is too strong and should be toned down to 'consistent with'. And similarly in line 153, the wording misleads when it says 'To test if GSK3 activity was elevated....' They are not 'testing' GSK3 activity but taking surrogate measurements. It might be better to start this paragraph – 'To look for further evidence that GSK3 activity was elevated...'

4. Lines 160-165. My reading of the data in Figure 2E is that the *sph/sph* CD4+ cells (without inhibitors) appear to be able to go through the same number of divisions as the WT cells after anti CD3+CD28 stimulation but have poor survival.

5. Line 167 and Fig. S4D. I think Fig. S4D should be omitted. I see no reason to re-visit the action of cyclosporin A when literature references cover the point being made.

6. Fig. 2F-H and lines 171-173. I welcome the additional explanation of 'similarity dilate' in the Methods section. I still suggest some improvement, however, on behalf of the reader. Panel 2F is essentially an illustration of the method being applied: one can stare at it for hours without it conveying much in terms of the Gimap5-related observation we are being told about. It is the

work that the 'similarity dilate' measure does that provides the 'data' here. I would suggest that the text or the figure legend here, i.e. at the point of describing the results, makes it clearer what the data in Fig 2G and H is all about.

7. Fig 2F – define BF.

8. Line 177 'accumulate protein expression' is an awkward, probably inaccurate, phrase.

9. Fig 3A. Explain time-course and nature of activation in legend.

10. Fig S5C: 'BDS' – why not in full as elsewhere?

11. Figure 3 legend: the hatched bars in panels H and I are not explained.

12. Line 239-240. I think that the reference to Figures here should perhaps read '(Fig. 3E-H)' (line 239) and '(Fig. 3E)' (line 240).

13. Line 242-243. 'Absence': we were told that P389-GSK3b is reduced, not absent.

14. Fig 5D legend should read 'Suppressive capacity OF regulatory T cells....'.

15. Line 324, Fig 8 compare line 198, 201, Fig. S5. Switch from use of 'Lamp1' terminology to CD107 terminology. Please make consistent for reader. Check also line 398.

16. Lines 340-1. I suggest better wording would be: 'failure to..... induce the nuclear translocation of TFs that is necessary for productive T cell proliferation and which occurs at an early stage.'

17. Line 342. Maybe I missed the evidence but why does this describe 'impaired nuclear translocation of P-Ser389 GSK3b', i.e. what is the evidence that it is the phospho-form that is being translocated? Why shouldn't it be the non-phospho form that is translocated and subsequently modified as part of intra-nuclear regulation? This is also relevant to Figure 9 and its legend.

18. Figure 9 legend: the text '2) the late stage nuclear translocation of P-Ser389 GSK3b required for the DNA damage response during cycling'. I thought that S389-phosphorylated GSK3b was an inactivated state while this text seems to suggest a positive 'requirement' for the phospho form rather than a requirement for GSK3b to be phosphorylated i.e. inactivated as part of the DNA damage response).

19. I find the red phosphorylation symbol next to the three GSK3 lozenges in the right hand panel of Figure 9 confusing.

20. Lines 344-5. I suggest better wording would be '...genetic deletion of GSK3b corrects T lymphocyte survival and prevents severe early-onset colitis....'

21. Lines 361-2. I suggest better wording would be 'More recently, phosphorylation of the Ser389 site in GSK3b has been implicated as a key step....'

22. Line 367. 'suggests'

23. Line 398. Lamp1. Is this your preferred nomenclature?

24. Line 447: 'Alzheimer's'

25. Line 471-2 'significance level'.

26. Line 479. Dr Hildeman is one of the authors so this statement is superfluous.

27. Line 487. 'Monoclonal'.

28. Line 496: 'medium'. Also line 558.

29. Line 546: '...spleen and lymph nodes OF WT and...'

30. Line 621: 'T cells WERE selectively expanded'.

31. Line 623 'use' better than 'using'.

32. Lines 634-5: better would be '....or involving chemical treatment of T cells...'

Reviewer #3 (Remarks to the Author):

The main conclusive experiments have simply not been done (even though asked for by the reviewers). With unconventional readout of Wnt signalling, nothing is seen, also for NFAT the experiments are somewhat more clear, but not fully conclusive. So how does GSK3b work? Cellular metabolism? DNA damage? Or still NFAT?

If authors cannot perform reporter assays, for Wnt, nuclear b-catenin levels could be checked, for NFAT translocation of NFAT to the nucleus etc

Reviewer #2 (Remarks to the Author):

We welcome this reviewer's thoughtful and meticulous comments. We have incorporated the suggested changes into the text. All textual changes in the revised manuscript are marked in red.

1. Lines 66-82. The new data on ex vivo survival of thymocytes and (later) peripheral T cells are valuable additions. However, (and I am not wishing to belabour this point) there are in these lines some inaccuracies in overstating the degree of consistency with the rat lyp/lyp model. Line 68 – there is in fact evidence against progressive loss of peripheral CD4+ cells in the rat model (see Yale J-F, Grose & Marliiss (1985) Diabetes 34:955: Time course of the lymphopenia in BB rats): lymphocyte numbers appear to start low and stay low. Similarly, the statement (line 80) that the new data on sph/sph RTEs 'are consistent with observations in the lyp/lyp rat' appears to overlook published data to the contrary (Zadeh et al. (1996) Autoimmunity 24:35; Ramanathan & Poussier (2001) Immunological Reviews 184:161). One has to conclude, in fact, that there are significant differences in detail between the Gimap5 sph/sph and the Gimap5 lyp/lyp models. Whether these differences are a matter of species difference or, alternatively, of the specific details of the genetic lesions under study, is so far unclear. There is a scatter of observations suggesting biological activity of fragments of Gimap5 (e.g. Sandal et al. (2003) Mol. Biol. Cell 14:3292). The three available mouse Gimap5 deficient strains and the rat model may differ in the persistence of cryptic levels of Gimap5 polypeptides with some activity.

We have changed the statements and removed "progressive" in line 67. We agree that there are slight differences in the phenotypes between the mouse models and rats that may be due to Gimap5 levels/fragments that may have residual activity. However, that is outside of the focus of the current manuscript.

2. Line 133. 'A marked reduction.....' Contrary to this statement, and after the updating of this figure, Fig 2 no longer shows a reduction of c-Myc in resting (unstimulated) sph/sph T cells. As one might expect, there is very little expression at all. Re-write required.

This has been corrected in the text and statement has been removed (line 132-133).

3. Line 150 – there is reference to resting T cells but Fig. S4A does not appear to show data for unstimulated cells, so this sentence should be modified. In the same sentence I think 'further indicating' is too strong and should be toned down to 'consistent with'. And similarly in line 153, the wording misleads when it says 'To test if GSK3 activity was elevated....' They are not 'testing' GSK3 activity but taking surrogate measurements. It might be better to start this paragraph – 'To look for further evidence that GSK3 activity was elevated...'

The suggested changes have been incorporated.

4. Lines 160-165. My reading of the data in Figure 2E is that the sph/sph CD4+ cells (without inhibitors) appear to be able to go through the same number of divisions as the WT cells after anti CD3+CD28 stimulation but have poor survival.

While some *Gimap5*^{sph/sph} CD4⁺ T cells undergo a similar number of divisions, even among surviving CD4⁺ T cells, the average number of proliferation cycles is reduced (Fig. 2E, row 2: α CD3/ α CD28).

5. Line 167 and Fig. S4D. I think Fig. S4D should be omitted. I see no reason to re-visit the action of cyclosporin A when literature references cover the point being made.

This experiment was requested by another reviewer.

6. Fig. 2F-H and lines 171-173. I welcome the additional explanation of ‘similarity dilate’ in the Methods section. I still suggest some improvement, however, on behalf of the reader. Panel 2F is essentially an illustration of the method being applied: one can stare at it for hours without it conveying much in terms of the Gimap5-related observation we are being told about. It is the work that the ‘similarity dilate’ measure does that provides the ‘data’ here. I would suggest that the text or the figure legend here, i.e. at the point of describing the results, makes it clearer what the data in Fig 2G and H is all about.

This issue has been clarified in the legend of Figure 2 and in addition in the text we included a reference to the method section (Line 167).

7. Fig 2F – define BF.

This has been clarified (legend Fig 2).

8. Line 177 ‘accumulate protein expression’ is an awkward, probably inaccurate, phrase.

This has been changed to “...a failure to accumulate TFs...” (Line 174-175)

9. Fig 3A. Explain time-course and nature of activation in legend.

This has been clarified in the legend of Fig.3.

10. Fig S5C: ‘BDS’ – why not in full as elsewhere?

This has been corrected.

11. Figure 3 legend: the hatched bars in panels H and I are not explained.

Hatched bars have now been clarified in the legend.

12. Line 239-240. I think that the reference to Figures here should perhaps read ‘(Fig. 3E-H)’ (line 239) and ‘(Fig. 3E)’ (line 240).

This has been corrected.

13. Line 242-243. ‘Absence’: we were told that P389-GSK3b is reduced, not absent.

This has been changed to “reduction” (Line 239)

14. Fig 5D legend should read ‘Suppressive capacity OF regulatory T cells....’.

Correction included.

15. Line 324, Fig 8 compare line 198, 201, Fig. S5. Switch from use of ‘Lamp1’ terminology to CD107 terminology. Please make consistent for reader. Check also line 398.

Lamp1 is used for mouse studies; CD107 is used for human.

16. Lines 340-1. I suggest better wording would be: ‘failure to..... induce the nuclear translocation of TFs that is necessary for productive T cell proliferation and which occurs at an early stage.’

We prefer “...that occurs at an early stage.”

17. Line 342. Maybe I missed the evidence but why does this describe ‘impaired nuclear translocation of P-Ser389 GSK3b’, i.e. what is the evidence that it is the phospho-form that is being translocated? Why shouldn’t it be the non-phospho form that is translocated and subsequently modified as part of intra-nuclear regulation? This is also relevant to Figure 9 and its legend.

The retention of pGSK3 β (Ser389) outside the nucleus in *Gimap5*^{sph/sph} CD4⁺ T cells (Fig. 3E-I) suggested to us that GSK3 β was phosphorylated before translocation. However, the reviewer is correct in that this is not definitively proven. As such, we have changed “nuclear translocation” to “nuclear accumulation” of pGSK3 β (Ser389).

18. Figure 9 legend: the text ‘2) the late stage nuclear translocation of P-Ser389 GSK3b required for the DNA damage response during cycling’. I thought that S389-phosphorylated GSK3b was an inactivated state while this text seems to suggest a positive ‘requirement’ for the phospho form rather than a requirement for GSK3b to be phosphorylated i.e. inactivated as part of the DNA damage response). We have reworded this section to clarify this point. It is important to note, however, that the role of GSK3(β) in the nucleus of activated CD4⁺ T cells remains undefined. Whether S389-phosphorylated GSK3 β plays an active role (despite being kinase-inactive), is also unknown.

19. I find the red phosphorylation symbol next to the three GSK3 lozenges in the right hand panel of Figure 9 confusing.

The phosphorylation was meant to convey that regulation of TFs is via GSK3 β -mediated phosphorylation leading to increased degradation or nuclear exclusion. Nonetheless, we have removed this symbol in Figure 9.

20. Lines 344-5. I suggest better wording would be ‘...genetic deletion of GSK3b corrects T lymphocyte survival and prevents severe early-onset colitis.....’

Alternative wording has been included as suggested.

21. Lines 361-2. I suggest better wording would be ‘More recently, phosphorylation of the Ser389 site in GSK3b has been implicated as a key step.....’

Alternative wording has been included as suggested.

22. Line 367. ‘suggests’

Corrected.

23. Line 398. Lamp1. Is this your preferred nomenclature?

Lamp1 is used for mouse studies; CD107 is used for human.

24. Line 447: ‘Alzheimer’s’

25. Line 471-2 ‘significance level’.

Corrections included.

26. Line 479. Dr Hildeman is one of the authors so this statement is superfluous.

Dr. Hildeman has been removed.

27. Line 487. ‘Monoclonal’.

28. Line 496: ‘medium’. Also line 558.

29. Line 546: ‘...spleen and lymph nodes OF WT and...’

30. Line 621: ‘T cells WERE selectively expanded’.
31. Line 623 ‘use’ better than ‘using’.
32. Lines 634-5: better would be ‘...or involving chemical treatment of T cells...’

All corrections have been included.

Reviewer #3 (Remarks to the Author):

The main conclusive experiments have simply not been done (even though asked for by the reviewers). With unconventional readout of Wnt signaling, nothing is seen, also for NFAT the experiments are somewhat more clear, but not fully conclusive. So how does GSK3b work? Cellular metabolism? DNA damage? Or still NFAT?

If authors cannot perform reporter assays, for Wnt, nuclear b-catenin levels could be checked, for NFAT translocation of NFAT to the nucleus etc.

We thank the reviewer for his/her interest in the paper. We would like to stress that our manuscript describes the regulatory role of *Gimap5* in GSK3 β inhibition during TCR signaling and not Wnt signaling. Our previous experiments using Wnt inhibitors (included in Sup. Fig.6) and our new studies directly assessing β -catenin and NFATc1 nuclear accumulation by Wnt3a, as suggested by the reviewer, fail to support a key role for Wnt signaling in GSK3 β control either by itself or during TCR signaling in CD4 $^+$ T cells. Specifically, Wnt3a stimulation of CD4 $^+$ T cells did not affect nuclear accumulation of unphosphorylated β -catenin (active), while α CD3/ α CD28 stimulation resulted in over a 2-fold increase in nuclear active β -catenin (Rebuttal Fig. 1A). Similarly, Wnt3a had limited effect on the accumulation of nuclear NFATc1, a process that requires calcium flux (Fig 1B-C).

Rebuttal Figure 1. Minimal effect of Wnt3a on nuclear accumulation of active β -catenin and NFATc1. (A) Nuclear expression of unphosphorylated β -catenin (Ser33/37/Thr41) in WT CD4 $^+$ T cells after stimulation with Wnt3a (200 ng/mL) \pm 2.5mL LiCl or α CD3/ α CD28 for 20h. (B) Nuclear expression of NFATc1 in CD4 $^+$ T cells after 4h resting or stimulation with Wnt3a. (C) ImageStream analysis of NFATc1 nuclear translocation in resting or Wnt3a-activated CD4 $^+$ T cells from WT or *Gimap5*^{spH/spH} mice. Negative ‘similarity’ values represent morphological separation between the two probes (NFATc1/7-AAD).

In addition, as previously demonstrated (Sup. Fig. 6), exogenous Wnt3a or Wnt blockade with IWP-2 has no effect on GSK3 β sequestration in CD4⁺ T cells nor affected their proliferation capacity, demonstrating that in primary CD4⁺ T cells, Wnt is neither sufficient nor necessary to induce GSK3 β sequestration and proliferation. Given the unique expression of Gimap5 in lymphocytes, we posit that it plays a key role in the TCR-induced regulation of GSK3 β , rather than signaling through the Wnt pathway.

As far as the functional consequences, our data clearly shows that GSK3 β is involved in several key pathways that include a transcriptional program as well as controlling the DNA damage response occurring during productive T cell proliferation. A failure to reduce GSK3 activity in the absence of Gimap5 affects all of these pathways, together leading to the devastating end stage phenotypes (impaired expansion, accumulation of DNA damage, cell death etc.). While dissecting the selective contribution of a single GSK3 target to these phenotypes is of interest, we believe this is outside of the scope of the current manuscript.